# A memory frontier for complex synapses

**Subhaneil Lahiri and Surya Ganguli**
Department of Applied Physics, Stanford University, Stanford CA
sulahiri@stanford.edu, sganguli@stanford.edu

## Abstract

An incredible gulf separates theoretical models of synapses, often described solely by a single scalar value denoting the size of a postsynaptic potential, from the immense complexity of molecular signaling pathways underlying real synapses. To understand the functional contribution of such molecular complexity to learning and memory, it is essential to expand our theoretical conception of a synapse from a single scalar to an entire dynamical system with many internal molecular functional states. Moreover, theoretical considerations alone demand such an expansion; network models with scalar synapses assuming finite numbers of distinguishable synaptic strengths have strikingly limited memory capacity. This raises the fundamental question, how does synaptic complexity give rise to memory? To address this, we develop new mathematical theorems elucidating the relationship between the structural organization and memory properties of complex synapses that are themselves molecular networks. Moreover, in proving such theorems, we uncover a framework, based on first passage time theory, to impose an order on the internal states of complex synaptic models, thereby simplifying the relationship between synaptic structure and function.

## 1  Introduction

It is widely thought that our very ability to remember the past over long time scales depends crucially on our ability to modify synapses in our brain in an experience dependent manner. Classical models of synaptic plasticity model synaptic efficacy as an analog scalar value, denoting the size of a postsynaptic potential injected into one neuron from another. Theoretical work has shown that such models have a reasonable, extensive memory capacity, in which the number of long term associations that can be stored by a neuron is proportional its number of afferent synapses [1–3]. However, recent experimental work has shown that many synapses are more digital than analog; they cannot robustly assume an infinite continuum of analog values, but rather can only take on a finite number of distinguishable strengths, a number than can be as small as two [4–6] (though see [7]). This one simple modification leads to a catastrophe in memory capacity: classical models with digital synapses, when operating in a palimpset mode in which the ongoing storage of new memories can overwrite previous memories, have a memory capacity proportional to the logarithm of the number of synapses [8, 9]. Intuitively, when synapses are digital, the storage of a new memory can flip a population of synaptic switches, thereby rapidly erasing previous memories stored in the same synaptic population. This result indicates that the dominant theoretical basis for the storage of long term memories in modifiable synaptic switches is flawed.

Recent work [10–12] has suggested that a way out of this logarithmic catastrophe is to expand our theoretical conception of a synapse from a single scalar value to an entire stochastic dynamical system in its own right. This conceptual expansion is further necessitated by the experimental reality that synapses contain within them immensely complex molecular signaling pathways, with many internal molecular functional states (e.g. see [4, 13, 14]). While externally, synaptic efficacy could be digital, candidate patterns of electrical activity leading to potentiation or depression could yield transitions between these internal molecular states without necessarily inducing an associated change in

synaptic efficacy. This form of synaptic change, known as metaplasticity [15, 16], can allow the probability of synaptic potentiation or depression to acquire a rich dependence on the history of prior changes in efficacy, thereby potentially improving memory capacity.

Theoretical studies of complex, metaplastic synapses have focused on analyzing the memory performance of a limited number of very specific molecular dynamical systems, characterized by a number of internal states in which potentiation and depression each induce a specific set of allowable transitions between states (e.g. see Figure 1 below). While these models can vastly outperform simple binary synaptic switches, these analyses leave open several deep and important questions. For example, how does the structure of a synaptic dynamical system determine its memory performance? What are the fundamental limits of memory performance over the space of all possible synaptic dynamical systems? What is the structural organization of synaptic dynamical systems that achieve these limits? Moreover, from an experimental perspective, it is unlikely that all synapses can be described by a single canonical synaptic model; just like the case of neurons, there is an incredible diversity of molecular networks underlying synapses both across species and across brain regions within a single organism [17]. In order to elucidate the functional contribution of this diverse molecular complexity to learning and memory, it is essential to move beyond the analysis of specific models and instead develop a general theory of learning and memory for complex synapses. Moreover, such a general theory of complex synapses could aid in development of novel artificial memory storage devices.

Here we initiate such a general theory by proving upper bounds on the memory curve associated with any synaptic dynamical system, within the well established ideal observer framework of [10, 11, 18]. Along the way we develop principles based on first passage time theory to order the structure of synaptic dynamical systems and relate this structure to memory performance. We summarize our main results in the discussion section.

## 2 Overall framework: synaptic models and their memory curves

In this section, we describe the class of models of synaptic plasticity that we are studying and how we quantify their memory performance. In the subsequent sections, we will find upper bounds on this performance.

We use a well established formalism for the study of learning and memory with complex synapses (see [10, 11, 18]). In this approach, electrical patterns of activity corresponding to candidate potentiating and depressing plasticity events occur randomly and independently at all synapses at a Poisson rate $r$. These events reflect possible synaptic changes due to either spontaneous network activity, or the storage of new memories. We let $f^{\text{pot}}$ and $f^{\text{dep}}$ denote the fraction of these events that are candidate potentiating or depressing events respectively. Furthermore, we assume our synaptic model has $M$ internal molecular functional states, and that a candidate potentiating (depotentiating) event induces a stochastic transition in the internal state described by an $M \times M$ discrete time Markov transition matrix $\mathbf{M}^{\text{pot}}$ ($\mathbf{M}^{\text{dep}}$). In this framework, the states of different synapses will be independent, and the entire synaptic population can be fully described by the probability distribution across these states, which we will indicate with the row-vector $\mathbf{p}(t)$. Thus the $i$'th component of $\mathbf{p}(t)$ denotes the fraction of the synaptic population in state $i$. Furthermore, each state $i$ has its own synaptic weight, $\mathbf{w}_i$, which we take, in the worst case scenario, to be restricted to two values. After shifting and scaling these two values, we can assume they are $\pm 1$, without loss of generality.

We also employ an "ideal observer" approach to the memory readout, where the synaptic weights are read directly. This provides an upper bound on the quality of any readout using neural activity.

For any single memory, stored at time $t = 0$, we assume there will be an ideal pattern of synaptic weights across a population of $N$ synapses, the $N$-element vector $\vec{w}_{\text{ideal}}$, that is $+1$ at all synapses that experience a candidate potentiation event, and $-1$ at all synapses that experience a candidate depression event at the time of memory storage. We assume that any pattern of synaptic weights close to $\vec{w}_{\text{ideal}}$ is sufficient to recall the memory. However, the actual pattern of synaptic weights at some later time, $t$, will change to $\vec{w}(t)$ due to further modifications from the storage of subsequent memories. We can use the overlap between these, $\vec{w}_{\text{ideal}} \cdot \vec{w}(t)$, as a measure of the quality of the memory. As $t \to \infty$, the system will return to its steady state distribution which will be uncorrelated

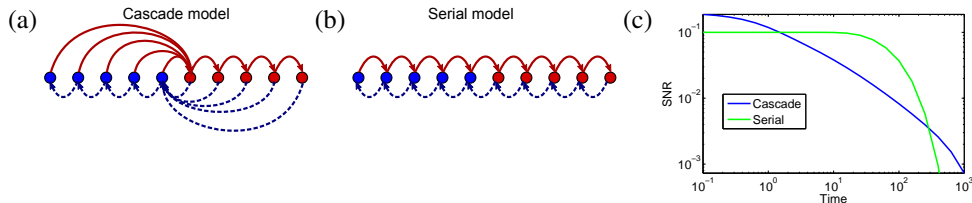

Figure 1: Models of complex synapses. (a) The cascade model of [10], showing transitions between states of high/low synaptic weight (red/blue circles) due to potentiation/depression (solid red/dashed blue arrows). (b) The serial model of [12]. (c) The memory curves of these two models, showing the decay of the signal-to-noise ratio (to be defined in §2) as subsequent memories are stored.

with the memory stored at $t = 0$. The probability distribution of the quantity $\vec{w}_{\text{ideal}} \cdot \vec{w}(\infty)$ can be used as a "null model" for comparison.

The extent to which the memory has been stored is described by a signal-to-noise ratio (SNR) [10, 11]:

$$\text{SNR}(t) = \frac{\langle \vec{w}_{\text{ideal}} \cdot \vec{w}(t) \rangle - \langle \vec{w}_{\text{ideal}} \cdot \vec{w}(\infty) \rangle}{\sqrt{\text{Var}(\vec{w}_{\text{ideal}} \cdot \vec{w}(\infty))}}. \tag{1}$$

The noise in the denominator is essentially $\sqrt{N}$. There is a correction when potentiation and depression are imbalanced, but this will not affect the upper bounds that we will discuss below and will be ignored in the subsequent formulae.

A simple average memory curve can be derived as follows. All of the preceding plasticity events, prior to $t = 0$, will put the population of synapses in its steady-state distribution, $\mathbf{p}^\infty$. The memory we are tracking at $t = 0$ will change the internal state distribution to $\mathbf{p}^\infty \mathbf{M}^{\text{pot}}$ (or $\mathbf{p}^\infty \mathbf{M}^{\text{dep}}$) in those synapses that experience a candidate potentiation (or depression) event. As the potentiating/depressing nature of the subsequent memories is independent of $\vec{w}_{\text{ideal}}$, we can average over all sequences, resulting in the evolution of the probability distribution:

$$\frac{d\mathbf{p}(t)}{dt} = r\mathbf{p}(t)\mathbf{W}^{\text{F}}, \qquad \text{where} \quad \mathbf{W}^{\text{F}} = f^{\text{pot}}\mathbf{M}^{\text{pot}} + f^{\text{dep}}\mathbf{M}^{\text{dep}} - \mathbf{I}. \tag{2}$$

Here $\mathbf{W}^{\text{F}}$ is a continuous time transition matrix that models the process of forgetting the memory stored at time $t = 0$ due to random candidate potentiation/depression events occurring at each synapse due to the storage of subsequent memories. Its stationary distribution is $\mathbf{p}^\infty$.

This results in the following SNR

$$\text{SNR}(t) = \sqrt{N} \left(2 f^{\text{pot}} f^{\text{dep}}\right) \mathbf{p}^\infty \left(\mathbf{M}^{\text{pot}} - \mathbf{M}^{\text{dep}}\right) e^{rt\mathbf{W}^{\text{F}}} \mathbf{w}. \tag{3}$$

A detailed derivation of this formula can be found in the supplementary material. We will frequently refer to this function as the memory curve. It can be thought of as the excess fraction of synapses (relative to equilibrium) that maintain their ideal synaptic strength at time $t$, as dictated by the stored memory at time $t = 0$.

Much of the previous work on these types of complex synaptic models has focused on understanding the memory curves of specific models, or choices of $\mathbf{M}^{\text{pot/dep}}$. Two examples of these models are shown in Figure 1. We see that they have different memory properties. The serial model performs relatively well at one particular timescale, but it performs poorly at other times. The cascade model does not perform quite as well at that time, but it maintains its performance over a wider range of timescales.

In this work, rather than analyzing specific models, we take a different approach, in order to obtain a more general theory. We consider the *entire* space of these models and find upper bounds on the memory capacity of *any* of them. The space of models with a fixed number of internal states $M$ is parameterized by the pair of $M \times M$ discrete time stochastic transition matrices $\mathbf{M}^{\text{pot}}$ and $\mathbf{M}^{\text{dep}}$, in addition to $f^{\text{pot/dep}}$. The parameters must satisfy the following constraints:

$$\mathbf{M}^{\text{pot/dep}}_{ij} \in [0, 1], \qquad f^{\text{pot/dep}} \in [0, 1], \qquad \mathbf{p}^\infty \mathbf{W}^{\text{F}} = 0, \qquad \mathbf{w}_i = \pm 1,$$

$$\sum_j \mathbf{M}^{\text{pot/dep}}_{ij} = 1, \qquad f^{\text{pot}} + f^{\text{dep}} = 1, \qquad \sum_i \mathbf{p}^\infty_i = 1. \tag{4}$$

The upper bounds on $\mathbf{M}_{ij}^{\text{pot/dep}}$ and $f^{\text{pot/dep}}$ follow automatically from the other constraints.

The critical question is: what do these constraints imply about the space of achievable memory curves in (3)? To answer this question, especially for limits on achievable memory at finite times, it will be useful to employ the eigenmode decomposition:

$$\mathbf{W}^{\text{F}} = \sum_a -q_a \mathbf{u}^a \mathbf{v}^a, \quad \mathbf{v}^a \mathbf{u}^b = \delta_{ab}, \quad \mathbf{W}^{\text{F}} \mathbf{u}^a = -q_a \mathbf{u}^a, \quad \mathbf{v}^a \mathbf{W}^{\text{F}} = -q_a \mathbf{v}^a. \quad (5)$$

Here $q_a$ are the negative of the eigenvalues of the forgetting process $\mathbf{W}^{\text{F}}$, $\mathbf{u}^a$ are the right (column) eigenvectors and $\mathbf{v}^a$ are the left (row) eigenvectors. This decomposition allows us to write the memory curve as a sum of exponentials,

$$\text{SNR}(t) = \sqrt{N} \sum_a \mathcal{I}_a e^{-rt/\tau_a}, \quad (6)$$

where $\mathcal{I}_a = (2f^{\text{pot}} f^{\text{dep}}) \mathbf{p}^\infty (\mathbf{M}^{\text{pot}} - \mathbf{M}^{\text{dep}}) \mathbf{u}^a \mathbf{v}^a \mathbf{w}$ and $\tau_a = 1/q_a$. We can then ask the question: what are the constraints on these quantities, namely eigenmode initial SNR's, $\mathcal{I}_a$, and time constants, $\tau_a$, implied by the constraints in (4)? We will derive some of these constraints in the next section.

# 3   Upper bounds on achievable memory capacity

In the previous section, in (3) we have described an analytic expression for a memory curve as a function of the structure of a synaptic dynamical system, described by the pair of stochastic transition matrices $\mathbf{M}^{\text{pot/dep}}$. Since the performance measure for memory is an entire memory curve, and not just a single number, there is no universal scalar notion of optimal memory in the space of synaptic dynamical systems. Instead there are tradeoffs between storing proximal and distal memories; often attempts to increase memory at late (early) times by changing $\mathbf{M}^{\text{pot/dep}}$, incurs a performance loss in memory at early (late) times in specific models considered so far [10–12]. Thus our end goal, achieved in §4, is to derive an envelope memory curve in the SNR-time plane, or a curve that forms an upper-bound on the *entire* memory curve of *any* model. In order to achieve this goal, in this section, we must first derive upper bounds, over the space of all possible synaptic models, on two different scalar functions of the memory curve: its initial SNR, and the area under the memory curve. In the process of upper-bounding the area, we will develop an essential framework to organize the structure of synaptic dynamical systems based on first passage time theory.

## 3.1   Bounding initial SNR

We now give an upper bound on the initial SNR,

$$\text{SNR}(0) = \sqrt{N} \left(2f^{\text{pot}} f^{\text{dep}}\right) \mathbf{p}^\infty \left(\mathbf{M}^{\text{pot}} - \mathbf{M}^{\text{dep}}\right) \mathbf{w}, \quad (7)$$

over *all* possible models and also find the class of models that saturate this bound. A useful quantity is the equilibrium probability flux between two disjoint sets of states, $\mathcal{A}$ and $\mathcal{B}$:

$$\mathbf{\Phi}_{\mathcal{A}\mathcal{B}} = \sum_{i \in \mathcal{A}} \sum_{j \in \mathcal{B}} r \mathbf{p}_i^\infty \mathbf{W}_{ij}^{\text{F}}. \quad (8)$$

The initial SNR is closely related to the flux from the states with $\mathbf{w}_i = -1$ to those with $\mathbf{w}_j = +1$ (see supplementary material):

$$\text{SNR}(0) \leq \frac{4\sqrt{N}\mathbf{\Phi}_{-+}}{r}. \quad (9)$$

This inequality becomes an equality if potentiation never decreases the synaptic weight and depression never increases it, which should be a property of any sensible model.

To maximize this flux, potentiation from a weak state must be guaranteed to end in a strong state, and depression must do the reverse. An example of such a model is shown in Figure 2(a,b). These models have a property known as "lumpability" (see [19, §6.3] for the discrete time version and [20, 21] for continuous time). They are completely equivalent (i.e. have the same memory curve) as a two state model with transition probabilities equal to 1, as shown in Figure 2(c).

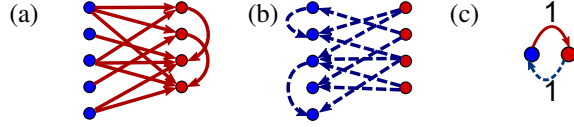

Figure 2: Synaptic models that maximize initial SNR. (a) For potentiation, all transitions starting from a weak state lead to a strong state, and the probabilities for all transitions leaving a given weak state sum to 1. (a) Depression is similar to potentiation, but with strong and weak interchanged. (c) The equivalent two state model, with transition probabilities under potentiation and depression equal to one.

This two state model has the equilibrium distribution $\mathbf{p}^\infty = (f^{\text{dep}}, f^{\text{pot}})$ and its flux is given by $\mathbf{\Phi}_{-+} = r f^{\text{pot}} f^{\text{dep}}$. This is maximized when $f^{\text{pot}} = f^{\text{dep}} = \frac{1}{2}$, leading to the upper bound:

$$\text{SNR}(0) \leq \sqrt{N}. \tag{10}$$

We note that while this model has high initial SNR, it also has very fast memory decay – with a timescale $\tau \sim \frac{1}{r}$. As the synapse is very plastic, the initial memory is encoded very easily, but the subsequent memories also overwrite it rapidly. This is one example of the tradeoff between optimizing memory at early versus late times.

## 3.2 Imposing order on internal states through first passage times

Our goal of understanding the relationship between structure and function in the space of all possible synaptic models is complicated by the fact that this space contains many different possible network topologies, encoded in the nonzero matrix elements of $\mathbf{M}^{\text{pot/dep}}$. To systematically analyze this entire space, we develop an important organizing principle using the theory of first passage times in the stochastic process of forgetting, described by $\mathbf{W}^{\text{F}}$. The mean first passage time matrix, $\overline{\mathbf{T}}_{ij}$, is defined as the average time it takes to reach state $j$ for the first time, starting from state $i$. The diagonal elements are defined to be zero.

A remarkable theorem we will exploit is that the quantity

$$\eta \equiv \sum_j \overline{\mathbf{T}}_{ij} \mathbf{p}_j^\infty, \tag{11}$$

known as Kemeny's constant (see [19, §4.4]), is independent of the starting state $i$. Intuitively, (11) states that the average time it takes to reach any state, weighted by its equilibrium probability, is independent of the starting state, implying a hidden constancy inherent in any stochastic process.

In the context of complex synapses, we can define the partial sums

$$\eta_i^+ = \sum_{j \in +} \overline{\mathbf{T}}_{ij} \mathbf{p}_j^\infty, \qquad \eta_i^- = \sum_{j \in -} \overline{\mathbf{T}}_{ij} \mathbf{p}_j^\infty. \tag{12}$$

These can be thought of as the average time it takes to reach the strong/weak states respectively. Using these definitions, we can then impose an order on the states by arranging them in order of decreasing $\eta_i^+$ or increasing $\eta_i^-$. Because $\eta_i^+ + \eta_i^- = \eta$ is independent of $i$, the two orderings are the same. In this order, which depends sensitively on the structure of $\mathbf{M}^{\text{pot/dep}}$, states later (to the right in figures below) can be considered to be more potentiated than states earlier (to the left in figures below), despite the fact that they have the same synaptic efficacy. In essence, in this order, a state is considered to be more potentiated if the average time it takes to reach all the strong efficacy states is shorter. We will see that synaptic models that optimize various measures of memory have an exceedingly simple structure when, and only when, their states are arranged in this order.[1]

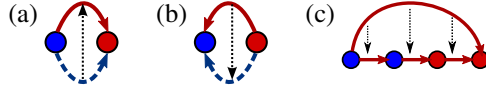

Figure 3: Perturbations that increase the area. (a) Perturbations that increase elements of $\mathbf{M}^{\text{pot}}$ above the diagonal and decrease the corresponding elements of $\mathbf{M}^{\text{dep}}$. It can no longer be used when $\mathbf{M}^{\text{dep}}$ is lower triangular, i.e. depression must move synapses to "more depressed" states. (b) Perturbations that decrease elements of $\mathbf{M}^{\text{pot}}$ below the diagonal and increase the corresponding elements of $\mathbf{M}^{\text{dep}}$. It can no longer be used when $\mathbf{M}^{\text{pot}}$ is upper triangular, i.e. potentiation must move synapses to "more potentiated" states. (c) Perturbation that decreases "shortcut" transitions and increases the bypassed "direct" transitions. It can no longer be used when there are only nearest-neighbor "direct" transitions.

## 3.3  Bounding area

Now consider the area under the memory curve:

$$A = \int_0^\infty \mathrm{d}t \ \text{SNR}(t). \tag{13}$$

We will find an upper bound on this quantity as well as the model that saturates this bound.

First passage time theory introduced in the previous section becomes useful because the area has a simple expression in terms of quantities introduced in (12) (see supplementary material):

$$
\begin{aligned}
A &= \sqrt{N}(4f^{\text{pot}}f^{\text{dep}}) \sum_{ij} \mathbf{p}_i^\infty \left( \mathbf{M}_{ij}^{\text{pot}} - \mathbf{M}_{ij}^{\text{dep}} \right) \left( \eta_i^+ - \eta_j^+ \right) \\
&= \sqrt{N}(4f^{\text{pot}}f^{\text{dep}}) \sum_{ij} \mathbf{p}_i^\infty \left( \mathbf{M}_{ij}^{\text{pot}} - \mathbf{M}_{ij}^{\text{dep}} \right) \left( \eta_j^- - \eta_i^- \right).
\end{aligned} \tag{14}
$$

With the states in the order described above, we can find perturbations of $\mathbf{M}^{\text{pot/dep}}$ that will always increase the area, whilst leaving the equilibrium distribution, $\mathbf{p}^\infty$, unchanged. Some of these perturbations are shown in Figure 3, see supplementary material for details. For example, in Figure 3(a), for two states $i$ on the left and $j$ on the right, with $j$ being more "potentiated" than $i$ (i.e. $\eta_i^+ > \eta_j^+$), we have proven that increasing $\mathbf{M}_{ij}^{\text{pot}}$ and decreasing $\mathbf{M}_{ij}^{\text{dep}}$ leads to an increase in area. The only thing that can prevent these perturbations from increasing the area is when they require the decrease of a matrix element that has already been set to 0. This determines the topology (non-zero transition probabilities) of the model with maximal area. It is of the form shown in Figure 4(c), with potentiation moving one step to the right and depression moving one step to the left. Any other topology would allow some class of perturbations (e.g. in Figure 3) to further increase the area.

As these perturbations do not change the equilibrium distribution, this means that the area of *any* model is bounded by that of a linear chain with the same equilibrium distribution. The area of a linear chain model can be expressed directly in terms of its equilibrium state distribution, $\mathbf{p}^\infty$, yielding the following upper bound on the area of any model with the same $\mathbf{p}^\infty$ (see supplementary material):

$$A \leq \frac{2\sqrt{N}}{r} \sum_k \left[ k - \sum_j j \mathbf{p}_j^\infty \right] \mathbf{p}_k^\infty \mathbf{w}_k = \frac{2\sqrt{N}}{r} \sum_k \left| k - \sum_j j \mathbf{p}_j^\infty \right| \mathbf{p}_k^\infty, \tag{15}$$

where we chose $\mathbf{w}_k = \text{sgn}[k - \sum_j j \mathbf{p}_j^\infty]$. We can then maximize this by pushing all of the equilibrium distribution symmetrically to the two end states. This can be done by reducing the transition probabilities out of these states, as in Figure 4(c). This makes it very difficult to exit these states once they have been entered. The resulting area is

$$A \leq \frac{\sqrt{N}(M-1)}{r}. \tag{16}$$

This analytical result is similar to a numerical result found in [18] under a slightly different information theoretic measure of memory performance.

The "sticky" end states result in very slow decay of memory, but they also make it difficult to encode the memory in the first place, since a small fraction of synapses are able to change synaptic efficacy during the storage of a new memory. Thus models that maximize area optimize memory at late times, at the expense of early times.

## 4 Memory curve envelope

Now we will look at the implications of the upper bounds found in the previous section for the SNR at finite times. As argued in (6), the memory curve can be written in the form

$$\text{SNR}(t) = \sqrt{N} \sum_a \mathcal{I}_a \mathrm{e}^{-rt/\tau_a}. \tag{17}$$

The upper bounds on the initial SNR, (10), and the area, (16), imply the following constraints on the parameters $\{\mathcal{I}_a, \tau_a\}$:

$$\sum_a \mathcal{I}_a \leq 1, \qquad \sum_a \mathcal{I}_a \tau_a \leq M - 1. \tag{18}$$

We are not claiming that these are a complete set of constraints: not every set $\{\mathcal{I}_a, \tau_a\}$ that satisfies these inequalities will actually be achievable by a synaptic model. However, any set that violates either inequality will definitely not be achievable.

Now we can pick some fixed time, $t_0$, and maximize the SNR at that time wrt. the parameters $\{\mathcal{I}_a, \tau_a\}$, subject to the constraints above. This always results in a single nonzero $\mathcal{I}_a$; in essence, optimizing memory at a single time requires a single exponential. The resulting optimal memory curve, along with the achieved memory at the chosen time, depends on $t_0$ as follows:

$$
\begin{aligned}
t_0 \leq \frac{M-1}{r} &\implies \text{SNR}(t) = \sqrt{N}\mathrm{e}^{-rt/(M-1)} &&\implies \text{SNR}(t_0) = \sqrt{N}\mathrm{e}^{-rt_0/(M-1)}, \\
t_0 \geq \frac{M-1}{r} &\implies \text{SNR}(t) = \frac{\sqrt{N}(M-1)\mathrm{e}^{-t/t_0}}{rt_0} &&\implies \text{SNR}(t_0) = \frac{\sqrt{N}(M-1)}{\mathrm{e}rt_0}.
\end{aligned}
\tag{19}
$$

Both the initial SNR bound and the area bound are saturated at early times. At late times, only the area bound is saturated. The function $\text{SNR}(t_0)$, the green curve in Figure 4(a), above forms a memory curve envelope with late-time power-law decay $\sim t_0^{-1}$. No synaptic model can have an SNR that is greater than this at any time. We can use this to find an upper bound on the memory lifetime, $\tau(\epsilon)$, by finding the point at which the envelope crosses $\epsilon$:

$$\tau(\epsilon) \leq \frac{\sqrt{N}(M-1)}{\epsilon \mathrm{e} r}, \tag{20}$$

where we assume $N > (\epsilon\mathrm{e})^2$. Intriguingly, both the lifetime and memory envelope expand linearly with the number of internal states $M$, and increase as the square root of the number of synapses $N$.

This leaves the question of whether this bound is achievable. At any time, can we find a model whose memory curve touches the envelope? The red curves in Figure 4(a) show the closest we have come to the envelope with actual models, by repeated numerical optimization of $\text{SNR}(t_0)$ over $\mathbf{M}^{\text{pot/dep}}$ with random initialization and by hand designed models.

We see that at early, but not late times, there is a gap between the upper bound that we can prove and what we can achieve with actual models. There may be other models we haven't found that could beat the ones we have, and come closer to our proven envelope. However, we suspect that the area constraint is not the bottleneck for optimizing memory at times less than $\mathcal{O}(\frac{M}{r})$. We believe there is some other constraint that prevents models from approaching the envelope, and currently are exploring several mathematical conjectures for the precise form of this constraint in order to obtain a potentially tighter envelope. Nevertheless, we have proven rigorously that no model's memory curve can ever exceed this envelope, and that it is at least tight for late times, longer than $\mathcal{O}(\frac{M}{r})$, where models of the form in Figure 4(c) can come close to the envelope.

## 5 Discussion

We have initiated the development of a general theory of learning and memory with complex synapses, allowing for an exploration of the entire space of complex synaptic models, rather than

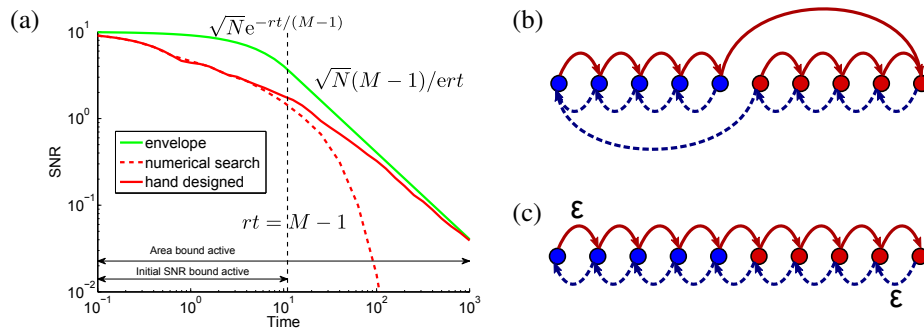

Figure 4: The memory curve envelope for $N = 100$, $M = 12$. (a) An upper bound on the SNR at any time is shown in green. The red dashed curve shows the result of numerical optimization of synaptic models with random initialization. The solid red curve shows the highest SNR we have found with hand designed models. At early times these models are of the form shown in (b) with different numbers of states, and all transition probabilities equal to 1. At late times they are of the form shown in (c) with different values of $\varepsilon$. The model shown in (c) also saturates the area bound (16) in the limit $\varepsilon \to 0$.

analyzing individual models one at a time. In doing so, we have obtained several new mathematical results delineating the functional limits of memory achievable by synaptic complexity, and the structural characterization of synaptic dynamical systems that achieve these limits. In particular, operating within the ideal observer framework of [10, 11, 18], we have shown that for a population of $N$ synapses with $M$ internal states, (a) the initial SNR of any synaptic model cannot exceed $\sqrt{N}$, and any model that achieves this bound is equivalent to a binary synapse, (b) the area under the memory curve of any model cannot exceed that of a linear chain model with the same equilibrium distribution, (c) both the area and memory lifetime of any model cannot exceed $\mathcal{O}(\sqrt{N}M)$, and the model that achieves this limit has a linear chain topology with only nearest neighbor transitions, (d) we have derived an envelope memory curve in the SNR-time plane that cannot be exceeded by the memory curve of any model, and models that approach this envelope for times greater than $\mathcal{O}(\frac{M}{r})$ are linear chain models, and (e) this late-time envelope is a power-law proportional to $\mathcal{O}(\sqrt{N}M/rt)$, indicating that synaptic complexity can strongly enhance the limits of achievable memory.

This theoretical study opens up several avenues for further inquiry. In particular, the tightness of our envelope for early times, less than $\mathcal{O}(\frac{M}{r})$, remains an open question, and we are currently pursuing several conjectures. We have also derived memory constrained envelopes, by asking in the space of models that achieve a given SNR at a given time, what is the maximal SNR achievable at other times. If these two times are beyond a threshold separation, optimal constrained models require two exponentials. It would be interesting to systematically analyze the space of models that achieve good memory at multiple times, and understand their structural organization, and how they give rise to multiple exponentials, leading to power law memory decays.

Finally, it would be interesting to design physiological experiments in order to perform optimal systems identification of potential Markovian dynamical systems hiding within biological synapses, given measurements of pre and post-synaptic spike trains along with changes in post-synaptic potentials. Then given our theory, we could match this measured synaptic model to optimal models to understand for which timescales of memory, if any, biological synaptic dynamics may be tuned.

In summary, we hope that a deeper theoretical understanding of the functional role of synaptic complexity, initiated here, will help advance our understanding of the neurobiology of learning and memory, aid in the design of engineered memory circuits, and lead to new mathematical theorems about stochastic processes.

**Acknowledgements**

We thank Sloan, Genenetech, Burroughs-Wellcome, and Swartz foundations for support. We thank Larry Abbott, Marcus Benna, Stefano Fusi, Jascha Sohl-Dickstein and David Sussillo for useful discussions.

## Footnotes

[1]Note that we do not need to worry about the order of the $\eta_i^\pm$ changing during the optimization: necessary conditions for a maximum only require that there is no infinitesimal perturbation that increases the area. Therefore we need only consider an infinitesimal neighborhood of the model, in which the order will not change.

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
