[Supplementary Material]

# A memory frontier for complex synapses: Supplementary material

**Subhaneil Lahiri and Surya Ganguli**
Department of Applied Physics, Stanford University, Stanford CA
sulahiri@stanford.edu, sulahiri@stanford.edu

Here we provide more details underlying the derivations of results in the main paper.

## 1 Continuous time Markov processes

In this section we'll provide a summary of all the relevant properties of ergodic Markov chains in continuous time to define notation. It is a generalization of material that can be found in [1] with some ideas from [2].

### 1.1 Notation

For any matrix $\mathbf{A}$, we define matrices $\mathbf{A}^{\text{dg}}$ and $\overline{\mathbf{A}}$ as

$$\mathbf{A}^{\text{dg}}_{ij} \equiv \delta_{ij}\mathbf{A}_{ij}, \qquad \overline{\mathbf{A}} \equiv \mathbf{A} - \mathbf{A}^{\text{dg}}. \tag{1}$$

We let $\mathbf{e}$ denote a column-vector of ones and $\mathbf{E} = \mathbf{e}\mathbf{e}^{\text{T}}$ denote a matrix of ones.

A continuous time Markov process is described by a matrix of transitions rates, $\mathbf{Q}_{ij}$, from state $i$ to $j$ with row sums equal to zero ($\mathbf{Q}\mathbf{e} = 0$). The probabilities of being in each state at time $t$, the row-vector $\mathbf{p}(t)$, evolve according to

$$\frac{\mathrm{d}\mathbf{p}(t)}{\mathrm{d}t} = \mathbf{p}(t)\mathbf{Q}, \tag{2}$$

where $\mathbf{p}(t)\mathbf{e} = 1$.

The equilibrium probabilities, $\mathbf{p}^{\infty}$, satisfy

$$\mathbf{p}^{\infty}\mathbf{Q} = 0, \qquad \mathbf{p}^{\infty}\mathbf{e} = 1. \tag{3}$$

As we assume an ergodic process, this eigenvalue is non-degenerate. If all other eigenvalues have strictly negative real parts, the process is regular (aperiodic).

We define additional matrices

$$\mathbf{\Lambda} \equiv (-\mathbf{Q}^{\text{dg}})^{-1}, \qquad \mathbf{P} \equiv \mathbf{I} + \mathbf{\Lambda}\mathbf{Q}. \tag{4}$$

It can be shown that $\mathbf{\Lambda}_{ii}$ is the mean time it takes to leave state $i$ and $\mathbf{P}_{ij}$ is the probability the the next transition from state $i$ goes to state $j$:

$$\mathbf{\Lambda}_{ii} = \frac{1}{\sum_{j \neq j}\mathbf{Q}_{ij}}, \qquad \mathbf{P}_{ij} = \begin{cases} 0 & \text{if } i = j, \\ \frac{\mathbf{Q}_{ij}}{\sum_{k \neq j}\mathbf{Q}_{ik}} & \text{otherwise.} \end{cases} \tag{5}$$

Furthermore, we also define

$$\mathbf{D} \equiv \mathrm{diag}(\mathbf{p}^{\infty})^{-1}, \qquad \implies \qquad \mathbf{p}^{\infty}\mathbf{D} = \mathbf{e}^{\text{T}}. \tag{6}$$

## 1.2 Fundamental matrix

For our results below regarding the integral of the memory curve, it can be useful to invert the stochastic transition matrix, $\mathbf{Q}$. However, since $\mathbf{Q}$ has a zero eigenvalue, it cannot be inverted. For this reason, the fundamental matrix arises as a useful surrogate for the inverse of $\mathbf{Q}$. It is related to the first passage times, as we will see in the next subsection. Here we define the fundamental matrix and review its properties.

**Definition 1: Fundamental matrix**
For discrete time, the generalized fundamental matrix was defined in [3]. For continuous time, we define:
$$\mathbf{Z} \equiv (-\mathbf{Q} + \mathbf{e}\boldsymbol{\pi})^{-1}, \tag{7}$$
where $\boldsymbol{\pi}$ is any row-vector with $\boldsymbol{\pi}\mathbf{e} = 1/\tau \neq 0$.

Note that the canonical choice for the discrete time version, $\boldsymbol{\pi} = \mathbf{p}^\infty$, is not available here due to problems with units. It will be helpful to choose $\boldsymbol{\pi}$ to be independent of $\mathbf{Q}$, e.g. $\boldsymbol{\pi} = \mathbf{e}^{\mathrm{T}}/(n\tau)$. All quantities that we calculate using $\mathbf{Z}$ below will be independent of this choice.

**Theorem 1:**
The definition of $\mathbf{Z}$ is valid, i.e. $(-\mathbf{Q} + \mathbf{e}\boldsymbol{\pi})$ is invertible.

*Proof.* Assume there exists an $\mathbf{x}$ such that
$$(-\mathbf{Q} + \mathbf{e}\boldsymbol{\pi})\mathbf{x} = 0. \tag{8}$$
Multiplying from the left with $\mathbf{p}^\infty$ gives
$$\boldsymbol{\pi}\mathbf{x} = 0. \tag{9}$$
Substituting back into (8) gives
$$\mathbf{Q}\mathbf{x} = 0.$$
As we assume an ergodic process, the zero eigenvalue is non-degenerate. Therefore, $\mathbf{x} = \lambda\mathbf{e}$. Substituting this into (9) gives
$$\lambda\boldsymbol{\pi}\mathbf{e} = \frac{\lambda}{\tau} = 0.$$
As we defined $\boldsymbol{\pi}$ such that $1/\tau \neq 0$, this means $\lambda = 0 \implies \mathbf{x} = 0.$ □

**Corollary 2:**

$$\boldsymbol{\pi}\mathbf{Z} = \mathbf{p}^\infty, \tag{10}$$
$$\mathbf{Z}\mathbf{e} = \tau\mathbf{e}, \tag{11}$$
$$\mathbf{I} + \mathbf{Q}\mathbf{Z} = \mathbf{e}\mathbf{p}^\infty, \tag{12}$$
$$\mathbf{I} + \mathbf{Z}\mathbf{Q} = \tau\mathbf{e}\boldsymbol{\pi}. \tag{13}$$

*Proof.* We can deduce (10) and (11) be pre/post-multiplying the following equations by $\mathbf{Z}$:
$$\mathbf{p}^\infty(-\mathbf{Q} + \mathbf{e}\boldsymbol{\pi}) = \boldsymbol{\pi},$$
$$(-\mathbf{Q} + \mathbf{e}\boldsymbol{\pi})\mathbf{e} = \frac{\mathbf{e}}{\tau}.$$
We can then deduce (12) and (13) by substituting these into
$$(-\mathbf{Q} + \mathbf{e}\boldsymbol{\pi})\mathbf{Z} = \mathbf{Z}(-\mathbf{Q} + \mathbf{e}\boldsymbol{\pi}) = \mathbf{I}.$$

□

## 1.3 First passage times

**Definition 2: First passage time matrix**
We define $\overline{\mathbf{T}}_{ij}$ as the mean time it takes the process to reach state $j$ for the first time, starting from state $i$. We also define $\mathbf{T}_{ii}^{\mathrm{dg}}$ as the mean time it takes the process to return to state $i$. As usual, $\mathbf{T} = \overline{\mathbf{T}} + \mathbf{T}^{\mathrm{dg}}$.

This matrix is given by

$$\mathbf{T} = (\mathbf{E}\mathbf{Z}^{\mathrm{dg}} - \mathbf{Z} + \mathbf{\Lambda})\mathbf{D}, \tag{14}$$

see [4] for a proof. We can separate this into its diagonal and off-diagonal pieces.

The recurrence times are given by

$$\mathbf{T}^{\mathrm{dg}} = \mathbf{\Lambda}\mathbf{D}. \tag{15}$$

or in component form

$$\mathbf{p}_i^\infty \mathbf{\Lambda}_{ii}^{-1} \mathbf{T}_{ii}^{\mathrm{dg}} = 1.$$

The extra factor of $\mathbf{\Lambda}_{ii}$, compared to the discrete case [1, Th.4.4.5], occurs because in this case we are demanding that the process leaves the initial state once before returning, whereas in the discrete case we only measure the time it takes to go to the initial state after the first time-step.

The off-diagonal mean first passage times are given by

$$\overline{\mathbf{T}} = (\mathbf{E}\mathbf{Z}^{\mathrm{dg}} - \mathbf{Z})\mathbf{D}. \tag{16}$$

or in component form:

$$\overline{\mathbf{T}}_{ij} = \frac{\mathbf{Z}_{jj} - \mathbf{Z}_{ij}}{\mathbf{p}_j^\infty}. \tag{17}$$

## 1.4  Mixing time (Kemeny's constant)

**Theorem 3:**
The quantity

$$\eta \equiv \sum_j \overline{\mathbf{T}}_{ij}\mathbf{p}_j^\infty \tag{18}$$

is independent of $i$.

*Proof.* For discrete time, a proof can be found in [1, Th.4.4.10]. For continuous time, we use (16), (11) and the transpose of (6):

$$\begin{aligned}
\overline{\mathbf{T}}(\mathbf{p}^\infty)^{\mathrm{T}} &= (\mathbf{E}\mathbf{Z}^{\mathrm{dg}} - \mathbf{Z})\mathbf{D}(\mathbf{p}^\infty)^{\mathrm{T}} \\
&= (\mathbf{e}\mathbf{e}^{\mathrm{T}}\mathbf{Z}^{\mathrm{dg}} - \mathbf{Z})\mathbf{e} \\
&= (\mathbf{e}^{\mathrm{T}}\mathbf{Z}^{\mathrm{dg}}\mathbf{e})\mathbf{e} - \mathbf{Z}\mathbf{e} \\
&= (\operatorname{tr}\mathbf{Z} - \tau)\mathbf{e}.
\end{aligned}$$

which proves (18) with $\eta = \operatorname{tr}\mathbf{Z} - \tau$. $\qquad\square$

Note that it is essential that we use $\overline{\mathbf{T}}$ and not $\mathbf{T}$ here, as that would lead to $\eta_i = \eta + \mathbf{\Lambda}_{ii}$, unlike the discrete time version, where this would only shift $\eta$ by 1.

## 1.5  Sensitivity of equilibrium distribution

Suppose that the Markov process, defined by $\mathbf{Q}$, depends on some parameter $\alpha$. Differentiating (7) gives

$$\frac{\mathrm{d}\mathbf{Z}}{\mathrm{d}\alpha} = \mathbf{Z}\frac{\mathrm{d}\mathbf{Q}}{\mathrm{d}\alpha}\mathbf{Z}. \tag{19}$$

We can substitute this into the derivative of (10):

$$\frac{\mathrm{d}\mathbf{p}^\infty}{\mathrm{d}\alpha} = \boldsymbol{\pi}\mathbf{Z}\frac{\mathrm{d}\mathbf{Q}}{\mathrm{d}\alpha}\mathbf{Z} = \mathbf{p}^\infty\frac{\mathrm{d}\mathbf{Q}}{\mathrm{d}\alpha}\mathbf{Z}. \tag{20}$$

We can rewrite this in component form and use the fact that $\mathbf{Q}_{ii} = -\sum_{i \neq j} \mathbf{Q}_{ij}$:

$$
\begin{aligned}
\frac{\mathrm{d}\mathbf{p}_k^\infty}{\mathrm{d}\alpha} &= \sum_{i,j} \mathbf{p}_i^\infty \frac{\mathrm{d}\mathbf{Q}_{ij}}{\mathrm{d}\alpha} \mathbf{Z}_{jk} \\
&= \sum_{i \neq j} \mathbf{p}_i^\infty \frac{\mathrm{d}\mathbf{Q}_{ij}}{\mathrm{d}\alpha} \mathbf{Z}_{jk} + \sum_i \mathbf{p}_i^\infty \frac{\mathrm{d}\mathbf{Q}_{ii}}{\mathrm{d}\alpha} \mathbf{Z}_{ik} \\
&= \sum_{i \neq j} \mathbf{p}_i^\infty \frac{\mathrm{d}\mathbf{Q}_{ij}}{\mathrm{d}\alpha} (\mathbf{Z}_{jk} - \mathbf{Z}_{ik}) \\
&= \sum_{i \neq j} \frac{\mathrm{d}\mathbf{Q}_{ij}}{\mathrm{d}\alpha} \mathbf{p}_i^\infty \mathbf{p}_k^\infty (\overline{\mathbf{T}}_{ik} - \overline{\mathbf{T}}_{jk}).
\end{aligned}
\tag{21}
$$

This is a generalization of a result of [5] from discrete to continuous time that we will need below. Note that the summand vanishes for $i = j$, so we can drop the restriction $i \neq j$ from the range of the sum.

## 1.6   Subsets and flux

Let us denote the set of states by $\mathcal{S}$. Consider a subset $\mathcal{A} \subset \mathcal{S}$. We can define a projection operator onto this subset:

$$
\left(\mathbf{I}^\mathcal{A}\right)_{ij} = \begin{cases} 1 & \text{if } i = j \in \mathcal{A}, \\ 0 & \text{otherwise.} \end{cases}
\tag{22}
$$

We will use superscripts/subscripts to denote projection onto/summation over a subset:

$$
\begin{aligned}
\boldsymbol{\pi}^\mathcal{A} = \boldsymbol{\pi}\mathbf{I}^\mathcal{A}, \quad \mathbf{M}^{\cdot\mathcal{A}} = \mathbf{M}\mathbf{I}^\mathcal{A}, \quad \mathbf{M}^{\mathcal{A}\cdot} = \mathbf{I}^\mathcal{A}\mathbf{M}, \qquad \mathbf{x}^\mathcal{A} = \mathbf{I}^\mathcal{A}\mathbf{x}, \\
\boldsymbol{\pi}_\mathcal{A} = \boldsymbol{\pi}\mathbf{e}^\mathcal{A}, \quad \mathbf{M}_{\cdot\mathcal{A}} = \mathbf{M}\mathbf{e}^\mathcal{A}, \quad \mathbf{M}_{\mathcal{A}\cdot} = \left(\mathbf{e}^\mathcal{A}\right)^\mathrm{T}\mathbf{M}, \quad \mathbf{x}_\mathcal{A} = \left(\mathbf{e}^\mathcal{A}\right)^\mathrm{T}\mathbf{x},
\end{aligned}
\tag{23}
$$

where $\boldsymbol{\pi}$ is a row vector, $\mathbf{M}$ is a matrix and $\mathbf{x}$ is a column vector.

We can define a flux matrix, a.k.a. ergodic flow:

$$
\boldsymbol{\Phi} = \mathbf{D}^{-1}\mathbf{Q}, \qquad \boldsymbol{\Phi}_{ij} = \mathbf{p}_i^\infty \mathbf{Q}_{ij}.
\tag{24}
$$

This measures the flow of probability between states in the equilibrium distribution. Detailed balance, a.k.a. reversibility, is equivalent to $\boldsymbol{\Phi} = \boldsymbol{\Phi}^\mathrm{T}$.

The flux between two subsets is a particularly useful quantity:

$$
\boldsymbol{\Phi}_{\mathcal{A}\mathcal{B}} = \mathbf{p}^{\infty\mathcal{A}}\mathbf{Q}\mathbf{e}^\mathcal{B}.
\tag{25}
$$

One can show that

$$
\boldsymbol{\Phi}_{\mathcal{A}\mathcal{A}^c} = \boldsymbol{\Phi}_{\mathcal{A}^c\mathcal{A}} = -\boldsymbol{\Phi}_{\mathcal{A}\mathcal{A}} = -\boldsymbol{\Phi}_{\mathcal{A}^c\mathcal{A}^c}
\tag{26}
$$

using $\left(\mathbf{p}^{\infty\mathcal{A}} + \mathbf{p}^{\infty\mathcal{A}^c}\right)\mathbf{Q} = 0$ and $\mathbf{Q}\left(\mathbf{e}^\mathcal{A} + \mathbf{e}^{\mathcal{A}^c}\right) = 0$.

## 1.7   Lumpability

Suppose we have partitioned the states into disjoint subsets, $\{\mathcal{A}_\alpha\}$:

$$
\bigcup_\alpha \mathcal{A}_\alpha = \mathcal{S}, \qquad \mathcal{A}_\alpha \cap \mathcal{A}_\beta = \delta_{\alpha\beta}\mathcal{A}_\alpha.
\tag{27}
$$

We will use $\alpha$ instead of $\mathcal{A}_\alpha$ in superscripts and subscripts in the following. The fact that these subsets are disjoint and exhaustive allows us to define the function

$$
\sigma(i) = \alpha \qquad \Longleftrightarrow \qquad i \in \mathcal{A}_\alpha.
\tag{28}
$$

We can use this partition to define a new stochastic process associated with the original Markov chain. At time $t$, if the state of the original process is $i$, the state of the new process is $\sigma(i)$.

One may ask if this new process is a Markov chain. The answer is yes, if the original Markov chain has a property called lumpability wrt. the partition (see [1, §6.3] for the discrete time version and [6, 7] for continuous time):

$$\sigma(i) = \sigma(j) \quad \Longrightarrow \quad \mathbf{Q}_{i\alpha} = \mathbf{Q}_{j\alpha} \equiv \sum_{k \in \mathcal{A}_\alpha} \mathbf{Q}_{jk}, \tag{29}$$

i.e. the total transition rate from some state to some subset is the same for all starting states within the same subset. This common value is the transition rate for the new lumped Markov chain.

This can be rewritten with the aid of two matrices

$$U_{\alpha i} = \frac{\delta_{\alpha\sigma(i)}}{|\mathcal{A}_\alpha|}, \qquad V_{i\alpha} = \delta_{\sigma(i)\alpha}. \tag{30}$$

Left multiplication by $U$ averages over subsets, right multiplication by $V$ sums over subsets. For $U$, we chose the uniform measure in each subset. Any measure would work equally well, e.g. one proportional to the equilibrium distribution:

$$U_{\alpha i} = \frac{\mathbf{p}_i^{\infty\,\alpha}}{\mathbf{p}_\alpha^\infty}. \tag{31}$$

One can show that $(UV)_{\alpha\beta} = \delta_{\alpha\beta}$. The matrix $VU$ is also interesting. It has a block diagonal structure, with each block corresponding to a subset. Each block is a discrete-time ergodic Markov matrix (it is an independent trials process with probabilities given by the measure chosen for $U$). This means that the right eigenvectors with eigenvalue 1 will be those that are constant in each subset:

$$VU\mathbf{x} = \mathbf{x} \quad \Longleftrightarrow \quad \mathbf{x} = \sum_\alpha x_\alpha \mathbf{e}^\alpha. \tag{32}$$

This allows us to write the lumpability condition (29), and the transition matrix for the lumped process compactly:

$$VU\mathbf{Q}V = \mathbf{Q}V, \qquad \widehat{\mathbf{Q}} = U\mathbf{Q}V. \tag{33}$$

By induction, one can show that similar relations hold for all powers:

$$VU\mathbf{Q}^n V = \mathbf{Q}^n V, \qquad \widehat{\mathbf{Q}}^n = U\mathbf{Q}^n V, \tag{34}$$

and, via the Taylor series, for the exponential as well:

$$VU\mathrm{e}^{t\mathbf{Q}}V = \mathrm{e}^{t\mathbf{Q}}V, \qquad \mathrm{e}^{t\widehat{\mathbf{Q}}} = U\mathrm{e}^{t\mathbf{Q}}V. \tag{35}$$

The equilibrium distribution of the lumped process is given by

$$\widehat{\mathbf{p}}^\infty = \mathbf{p}^\infty V. \tag{36}$$

## 2 Signal-to-Noise ratio (SNR)

In this section we will look at the signal-to-noise curve, and put an upper bound on its initial value. We need only consider ergodic Markov chains. Transient states would be unoccupied in equilibrium and would not be accessed by the signal creation process, therefore they could be removed from the analysis. Absorbing chains are degenerate cases: they have zero initial signal but infinite decay times, so they can only be approached as the limit of a sequence of ergodic chains.

### 2.1 Framework

The individual potentiation/depression events will be described by *discrete*-time Markov chains:

$$\mathbf{M}^{\mathrm{pot/dep}} \equiv \mathbf{I} + \mathbf{W}^{\mathrm{pot/dep}}, \qquad \mathbf{M}^{\mathrm{pot/dep}}\mathbf{e} = \mathbf{e}, \qquad \mathbf{M}_{ij}^{\mathrm{pot/dep}} \in [0, 1]. \tag{37}$$

The initial signal creation event occurs at time $t = 0$, but all subsequent potentiation/depression events occur at random times according to Poisson processes with rates $rf^{\mathrm{pot/dep}}$, where $f^{\mathrm{pot}} + f^{\mathrm{dep}} =$

1 are the fraction of plasticity events that are potentiating/depressing respectively. This means that the "forgetting" process will be described by the *continuous*-time Markov chain:

$$\mathbf{Q} = r\mathbf{W}^{\mathrm{F}} \equiv r\left(f^{\mathrm{pot}}\mathbf{W}^{\mathrm{pot}} + f^{\mathrm{dep}}\mathbf{W}^{\mathrm{dep}}\right). \tag{38}$$

We only require that this Markov chain is ergodic. The Markov chains described by $\mathbf{M}^{\mathrm{pot/dep}}$ need not be.

We assume that the probability distribution starts in the equilibrium distribution (3). During the initial signal creation, a fraction $f^{\mathrm{pot}}$ will change to $\mathbf{p}^\infty\mathbf{M}^{\mathrm{pot}}$ and a fraction $f^{\mathrm{dep}}$ will change to $\mathbf{p}^\infty\mathbf{M}^{\mathrm{dep}}$. After this, probabilities will evolve according to (2).

## 2.2 SNR curve

As discussed in the main text, the signal-to-noise ratio is given by

$$\mathrm{SNR}(t) = \frac{\langle \vec{w}_{\mathrm{ideal}} \cdot \vec{w}(t)\rangle - \langle \vec{w}_{\mathrm{ideal}} \cdot \vec{w}(\infty)\rangle}{\sqrt{\mathrm{Var}(\vec{w}_{\mathrm{ideal}} \cdot \vec{w}(\infty))}}. \tag{39}$$

First, let's look at the denominator, remembering that the states and plasticity events of each synapse are independent and identically distributed:

$$
\begin{aligned}
\mathrm{Var}(\vec{w}_{\mathrm{ideal}} \cdot \vec{w}(\infty)) &= \sum_{\alpha\beta} \left\langle \vec{w}^{\alpha}_{\mathrm{ideal}}\vec{w}^{\alpha}(\infty)\vec{w}^{\beta}_{\mathrm{ideal}}\vec{w}^{\beta}(\infty)\right\rangle - \left(\sum_{\alpha}\langle \vec{w}^{\alpha}_{\mathrm{ideal}}\vec{w}^{\alpha}(\infty)\rangle\right)^2 \\
&= \sum_{\alpha}\left\langle (\vec{w}^{\alpha}_{\mathrm{ideal}})^2(\vec{w}^{\alpha}(\infty))^2\right\rangle + \sum_{\alpha\neq\beta}\langle \vec{w}^{\alpha}_{\mathrm{ideal}}\vec{w}^{\alpha}(\infty)\rangle\langle \vec{w}^{\beta}_{\mathrm{ideal}}\vec{w}^{\beta}(\infty)\rangle \\
&\qquad - \left(\sum_{\alpha}\langle \vec{w}^{\alpha}_{\mathrm{ideal}}\vec{w}^{\alpha}(\infty)\rangle\right)^2 \\
&= N\langle 1\rangle + N(N-1)\left\langle \vec{w}^{1}_{\mathrm{ideal}}\vec{w}^{1}(\infty)\right\rangle^2 - N^2\left\langle \vec{w}^{1}_{\mathrm{ideal}}\vec{w}^{1}(\infty)\right\rangle^2 \\
&= N(1 - \left\langle \vec{w}^{1}_{\mathrm{ideal}}\vec{w}^{1}(\infty)\right\rangle^2),
\end{aligned}
\tag{40}
$$

where we used $\vec{w}^{\alpha} = \pm 1$.

For the numerator, we can write

$$\langle \vec{w}_{\mathrm{ideal}} \cdot \vec{w}(t)\rangle = \sum_{\alpha}\langle \vec{w}^{\alpha}_{\mathrm{ideal}}\vec{w}^{\alpha}(t)\rangle = N\left\langle \vec{w}^{1}_{\mathrm{ideal}}\vec{w}^{1}(t)\right\rangle, \tag{41}$$

Noting that $\vec{w}_{\mathrm{ideal}} = \pm 1$ with probability $f^{\mathrm{pot/dep}}$,

$$
\begin{aligned}
\left\langle \vec{w}^{1}_{\mathrm{ideal}}\vec{w}^{1}(t)\right\rangle &= f^{\mathrm{pot}}\left\langle \vec{w}^{1}(t)\right\rangle_{\mathrm{pot},t=0} - f^{\mathrm{dep}}\left\langle \vec{w}^{1}(t)\right\rangle_{\mathrm{dep},t=0} \\
&= f^{\mathrm{pot}}\sum_{i}P(\mathrm{state}=i,t\mid\mathrm{pot},0)\mathbf{w}_i - f^{\mathrm{dep}}\sum_{i}P(\mathrm{state}=i,t\mid\mathrm{dep},0)\mathbf{w}_i.
\end{aligned}
\tag{42}
$$

From the previous section,

$$P(\mathrm{state}=i,t\mid\mathrm{pot/dep},0) = \left[\mathbf{p}^\infty\mathbf{M}^{\mathrm{pot/dep}}\,\mathrm{e}^{rt\mathbf{W}^{\mathrm{F}}}\right]_i, \tag{43}$$

which describes the synapses starting in the equilibrium distribution, changing state due to the plasticity event at $t=0$ and subsequent evolution according to (2) due to plasticity events uncorrelated

with $\vec{w}_{\text{ideal}}$.[1] This results in

$$\left\langle \vec{w}_{\text{ideal}}^1 \vec{w}^1(t) \right\rangle = \mathbf{p}^\infty (f^{\text{pot}} \mathbf{M}^{\text{pot}} - f^{\text{dep}} \mathbf{M}^{\text{dep}}) \, \mathrm{e}^{rt\mathbf{W}^{\text{F}}} \mathbf{w},$$
$$\left\langle \vec{w}_{\text{ideal}}^1 \vec{w}^1(\infty) \right\rangle = \mathbf{p}^\infty (f^{\text{pot}} \mathbf{M}^{\text{pot}} - f^{\text{dep}} \mathbf{M}^{\text{dep}}) \, \mathbf{e} \mathbf{p}^\infty \mathbf{w}$$
$$= \mathbf{p}^\infty (f^{\text{pot}} \mathbf{e} - f^{\text{dep}} \mathbf{e}) \, \mathbf{p}^\infty \mathbf{w} \qquad (44)$$
$$= (f^{\text{pot}} - f^{\text{dep}}) \, \mathbf{p}^\infty \mathbf{w}$$
$$= (f^{\text{pot}} - f^{\text{dep}}) \, \mathbf{p}^\infty \mathrm{e}^{rt\mathbf{W}^{\text{F}}} \mathbf{w}.$$

Combining these allows us to write the numerator as

$$\langle \vec{w}_{\text{ideal}} \cdot \vec{w}(t) \rangle - \langle \vec{w}_{\text{ideal}} \cdot \vec{w}(\infty) \rangle = N\mathbf{p}^\infty (f^{\text{pot}}(\mathbf{M}^{\text{pot}} - \mathbf{I}) - f^{\text{dep}}(\mathbf{M}^{\text{dep}} - \mathbf{I})) \, \mathrm{e}^{rt\mathbf{W}^{\text{F}}} \mathbf{w}$$
$$= N\mathbf{p}^\infty (f^{\text{pot}}(\mathbf{W}^{\text{pot}} - \mathbf{W}^{\text{F}}) - f^{\text{dep}}(\mathbf{W}^{\text{dep}} - \mathbf{W}^{\text{F}})) \, \mathrm{e}^{rt\mathbf{W}^{\text{F}}} \mathbf{w} \quad (45)$$
$$= N(2f^{\text{pot}} f^{\text{dep}})\mathbf{p}^\infty (\mathbf{W}^{\text{pot}} - \mathbf{W}^{\text{dep}}) \, \mathrm{e}^{rt\mathbf{W}^{\text{F}}} \mathbf{w}.$$

where we used $\mathbf{p}^\infty \mathbf{W}^{\text{F}} = 0$ in going from the first to second lines. Combining with (40) gives

$$\mathrm{SNR}(t) = \frac{\sqrt{N}(2f^{\text{pot}} f^{\text{dep}})\mathbf{p}^\infty (\mathbf{W}^{\text{pot}} - \mathbf{W}^{\text{dep}}) \, \mathrm{e}^{rt\mathbf{W}^{\text{F}}} \mathbf{w}}{\sqrt{1 - (f^{\text{pot}} - f^{\text{dep}})^2 (\mathbf{p}_+^\infty - \mathbf{p}_-^\infty)^2}}. \qquad (46)$$

The denominator will not play any role in what follows, as the models that maximize the various measures of memory performance all have some sort of balance between potentiation and depression, either with $f^{\text{pot}} = f^{\text{dep}}$ or $\mathbf{p}_+^\infty = \mathbf{p}_-^\infty$. We can set the denominator to 1 without changing any of our results.

This results in our final formula:

$$\mathrm{SNR}(t) = \sqrt{N}(2f^{\text{pot}} f^{\text{dep}}) \, \mathbf{p}^\infty (\mathbf{W}^{\text{pot}} - \mathbf{W}^{\text{dep}}) \, \mathrm{e}^{rt\mathbf{W}^{\text{F}}} \mathbf{w}. \qquad (47)$$

The factor of $\mathbf{p}^\infty$ describes the synapses being in the steady-state distribution before the memory is encoded. The factor of $(\mathbf{M}^{\text{pot}} - \mathbf{M}^{\text{dep}})$ comes from the encoding of the memory at $t = 0$, with $\vec{w}_{\text{ideal}}$ being $\pm 1$ in synapses that are potentiated/depotentiated. The factor of $\mathrm{e}^{rt\mathbf{W}^{\text{F}}}$ describes the subsequent evolution of the probability distribution, averaged over all sequences of plasticity events, and the factor of $\mathbf{w}$ indicates the readout via the synaptic weight.

We can express this in terms of the one parameter family of transition matrices:

$$\mathbf{W}(\alpha) = \alpha \mathbf{W}^{\text{pot}} + (1 - \alpha)\mathbf{W}^{\text{dep}}, \qquad \Longrightarrow \qquad \mathbf{W}^{\text{F}} = \mathbf{W}(f^{\text{pot}}),$$
$$\mathbf{W}^{\text{pot}} - \mathbf{W}^{\text{dep}} = \frac{\mathrm{d}\mathbf{W}}{\mathrm{d}\alpha}, \qquad (48)$$
$$\mathbf{p}^\infty \frac{\mathrm{d}\mathbf{W}}{\mathrm{d}\alpha} = -\frac{\mathrm{d}\mathbf{p}^\infty}{\mathrm{d}\alpha} \mathbf{W}^{\text{F}}.$$

Then (47) becomes

$$\mathrm{SNR}(t) = \sqrt{N}(2f^{\text{pot}} f^{\text{dep}}) \frac{\mathrm{d}\mathbf{p}^\infty}{\mathrm{d}\alpha}(-\mathbf{W}^{\text{F}}) \, \mathrm{e}^{rt\mathbf{W}^{\text{F}}} \mathbf{w}. \qquad (49)$$

### 2.3 Lumpability and the SNR curve

Suppose that we have a partition such that $\mathbf{W}^{\text{pot}}$ and $\mathbf{W}^{\text{dep}}$ are simultaneously lumpable, and that all the states in each subset have the same synaptic strength (see §1.7):

$$VU\mathbf{W}^{\text{pot/dep}}V = \mathbf{W}^{\text{pot/dep}}V, \qquad VU\mathbf{w} = \mathbf{w}. \qquad (50)$$

$$\mathrm{e}^{rt\mathbf{W}^{\text{F}}} = \sum_{n=0}^\infty \frac{(rt)^n \, \mathrm{e}^{-rt}}{n!} \sum_{m=0}^n (f^{\text{pot}})^m (f^{\text{dep}})^{n-m} \left[ \mathbf{M}^{\text{pot}} \mathbf{M}^{\text{dep}} \mathbf{M}^{\text{pot}} \mathbf{M}^{\text{pot}} \dots + \text{permutations} \right].$$

Thus, evolving according to (2) results in averaging over all sequences of plasticity events, as we only need linear expectations of $\vec{w}(t)$ in the end.

We can define a new synapse with

$$\widehat{\mathbf{W}}^{\text{pot/dep}} = U\mathbf{W}^{\text{pot/dep}}V, \qquad \widehat{\mathbf{w}} = U\mathbf{w}, \qquad \widehat{\mathbf{p}}^\infty = \mathbf{p}^\infty V. \tag{51}$$

This synapse has an SNR curve:

$$
\begin{aligned}
\frac{\text{SNR}(t)}{\sqrt{N}(2f^{\text{pot}}f^{\text{dep}})} &= \widehat{\mathbf{p}}^\infty(\widehat{\mathbf{W}}^{\text{pot}} - \widehat{\mathbf{W}}^{\text{dep}})\mathrm{e}^{rt\widehat{\mathbf{W}}^F}\widehat{\mathbf{w}}. \\
&= \mathbf{p}^\infty VU(\mathbf{W}^{\text{pot}} - \mathbf{W}^{\text{dep}})VU\mathrm{e}^{rt\mathbf{W}^F}VU\mathbf{w}. \\
&= \mathbf{p}^\infty(\mathbf{W}^{\text{pot}} - \mathbf{W}^{\text{dep}})VU\mathrm{e}^{rt\mathbf{W}^F}VU\mathbf{w}. \\
&= \mathbf{p}^\infty(\mathbf{W}^{\text{pot}} - \mathbf{W}^{\text{dep}})\mathrm{e}^{rt\mathbf{W}^F}VU\mathbf{w}. \\
&= \mathbf{p}^\infty(\mathbf{W}^{\text{pot}} - \mathbf{W}^{\text{dep}})\mathrm{e}^{rt\mathbf{W}^F}\mathbf{w}.
\end{aligned}
\tag{52}
$$

i.e. the lumped process has exactly the same SNR as the original one.

## 2.4 Initial SNR and flux

Using $\mathbf{p}^\infty\mathbf{W}^F = 0$ and the first line of (45), we can write the initial SNR as

$$\frac{\text{SNR}(0)}{\sqrt{N}} = I = (\mathbf{p}^{\infty+} + \mathbf{p}^{\infty-})(f^{\text{pot}}\mathbf{W}^{\text{pot}} - f^{\text{dep}}\mathbf{W}^{\text{dep}})(\mathbf{e}^+ - \mathbf{e}^-). \tag{53}$$

Using $\mathbf{W}^{\text{pot/dep}}(\mathbf{e}^+ + \mathbf{e}^-) = 0$ and (26):

$$r\mathbf{p}^{\infty-}(f^{\text{pot}}\mathbf{W}^{\text{pot}} + f^{\text{dep}}\mathbf{W}^{\text{dep}})\mathbf{e}^+ = \mathbf{\Phi}_{-+} = \mathbf{\Phi}_{+-} = r\mathbf{p}^{\infty+}(f^{\text{pot}}\mathbf{W}^{\text{pot}} + f^{\text{dep}}\mathbf{W}^{\text{dep}})\mathbf{e}^-,$$

we can rewrite (53) as

$$I = \frac{4\mathbf{\Phi}_{-+}}{r} - 4\mathbf{p}^{\infty+}f^{\text{pot}}\mathbf{W}^{\text{pot}}\mathbf{e}^- - 4\mathbf{p}^{\infty-}f^{\text{dep}}\mathbf{W}^{\text{dep}}\mathbf{e}^+. \tag{54}$$

The last two terms are guaranteed to be negative, as the diagonal parts of $\mathbf{W}^{\text{pot/dep}}$ cannot contribute. Therefore

$$\text{SNR}(0) \leq \frac{4\sqrt{N}\mathbf{\Phi}_{-+}}{r}. \tag{55}$$

This inequality is saturated if potentiation never takes it from a $+$ state to a $-$ state and depression never takes it from a $-$ state to a $+$ state.

# 3 Area maximisation

In this section we will find an upper bound on the area under the signal-to-noise curve. As in §2, we will only consider ergodic Markov chains. We will see in §3.4 that the optimal chain is absorbing, so it lies on the boundary of the (open) set of ergodic chains, but it still puts an upper bound on the area for any chain in the interior.

## 3.1 Area under signal-to-noise curve

The signal-to-noise curve is given by (49). The area is computed by integrating this

$$
\begin{aligned}
A &= \frac{\sqrt{N}(2f^{\text{pot}}f^{\text{dep}})}{r}\frac{\mathrm{d}\mathbf{p}^\infty}{\mathrm{d}\alpha}\left[-\mathrm{e}^{rt\mathbf{W}^F}\right]_0^\infty\mathbf{w} \\
&= \frac{\sqrt{N}(2f^{\text{pot}}f^{\text{dep}})}{r}\frac{\mathrm{d}\mathbf{p}^\infty}{\mathrm{d}\alpha}(\mathbf{I} - \mathbf{e}\mathbf{p}^\infty)\mathbf{w} \\
&= \frac{\sqrt{N}(2f^{\text{pot}}f^{\text{dep}})}{r}\frac{\mathrm{d}\mathbf{p}^\infty}{\mathrm{d}\alpha}\mathbf{w}.
\end{aligned}
\tag{56}
$$

We can rewrite this using (21), with $A = \sqrt{N}(2f^{\text{pot}}f^{\text{dep}})\hat{A}$ and $\mathbf{q}_{ij} \equiv \frac{\mathrm{d}\mathbf{W}^F_{ij}}{\mathrm{d}\alpha} = \mathbf{W}^{\text{pot}}_{ij} - \mathbf{W}^{\text{dep}}_{ij}$

$$\hat{A} = \sum_{i,j,k}\mathbf{p}_i^\infty\mathbf{q}_{ij}(\overline{\mathbf{T}}_{ik} - \overline{\mathbf{T}}_{jk})\mathbf{p}_k^\infty\mathbf{w}_k. \tag{57}$$

**Definition 3: Partial mixing times**
We define the $\pm$ mixing times as

$$
\begin{aligned}
\eta_i^\pm \equiv \sum_k \overline{\mathbf{T}}_{ik}\mathbf{p}_k^\infty \left(\frac{1 \pm \mathbf{w}_k}{2}\right) \quad &= \sum_{k \in \pm} \overline{\mathbf{T}}_{ik}\mathbf{p}_k^\infty \\
= \sum_k (\mathbf{Z}_{kk} - \mathbf{Z}_{ik})\left(\frac{1 \pm \mathbf{w}_k}{2}\right) \quad &= \sum_{k \in \pm} (\mathbf{Z}_{kk} - \mathbf{Z}_{ik}).
\end{aligned}
\tag{58}
$$

We can think of $\eta_i^+$ as a measure of the "distance" to the $\mathbf{w}_k = +1$ states and $\eta_i^-$ as the "distance" to the $\mathbf{w}_k = -1$ states.

Using (18), we can write:

$$
\begin{aligned}
\eta_i^+ + \eta_i^- &= \eta, \\
2(\eta_i^+ - \eta_j^+) &= \sum_k (\overline{\mathbf{T}}_{ik} - \overline{\mathbf{T}}_{jk})\mathbf{p}_k^\infty \mathbf{w}_k = \sum_k (\mathbf{Z}_{jk} - \mathbf{Z}_{ik})\mathbf{w}_k.
\end{aligned}
\tag{59}
$$

We could arrange the states in order of decreasing $\eta^+$, which is the same as the order of increasing $\eta^-$.

We can rewrite (57) as

$$
\begin{aligned}
\hat{A} = 2\sum_{i,j} \mathbf{q}_{ij}\mathbf{p}_i^\infty (\eta_i^+ - \eta_j^+) \quad &= -2\sum_{i,j} \mathbf{q}_{ij}\mathbf{p}_i^\infty \eta_j^+ \\
= 2\sum_{i,j} \mathbf{q}_{ij}\mathbf{p}_i^\infty (\eta_j^- - \eta_i^-) \quad &= 2\sum_{i,j} \mathbf{q}_{ij}\mathbf{p}_i^\infty \eta_j^-.
\end{aligned}
\tag{60}
$$

We can also express it in terms of the fundamental matrix (7) as

$$
\hat{A} = \sum_{i,j,k,l} \mathbf{q}_{ij}\boldsymbol{\pi}_l \mathbf{Z}_{li}(\mathbf{Z}_{jk} - \mathbf{Z}_{ik})\mathbf{w}_k = \boldsymbol{\pi}\mathbf{Z}q\mathbf{Z}\mathbf{w}.
\tag{61}
$$

It is also helpful to define the following quantities:

$$
\begin{aligned}
c_k &= \frac{d \ln \mathbf{p}_k^\infty}{d\alpha} = \sum_{ij} \mathbf{p}_i^\infty \mathbf{q}_{ij}\left(\overline{\mathbf{T}}_{ik} - \overline{\mathbf{T}}_{jk}\right) = -\left(\mathbf{p}^\infty q\overline{\mathbf{T}}\right)_k = \frac{(\mathbf{p}^\infty q\mathbf{Z})_k}{\mathbf{p}_k^\infty}, \\
a_i &= \sum_j \mathbf{q}_{ij}\mathbf{p}_i^\infty (\eta_i^+ - \eta_j^+), \\
\implies \hat{A} &= \sum_k c_k \mathbf{p}_k^\infty \mathbf{w}_k = 2\sum_i a_i.
\end{aligned}
\tag{62}
$$

Note that the optimal choice of $\mathbf{w}$ is $\mathbf{w}_k = \operatorname{sgn}(c_k)$.

## 3.2 Derivatives wrt. $\mathbf{W}^{\text{pot/dep}}$

In the following, we will mathematically define the classes of perturbations pictorially described in Figure 3 of the main paper. In order to do so, we will need to consider expressions for derivatives of various quantities with respect to $\mathbf{W}_{ij}^{\text{pot/dep}}$.

As discussed in the main text, we will regard the off-diagonal elements of $\mathbf{W}_{ij}^{\text{pot/dep}}$ to be the independent variables, with $\mathbf{W}_{ii}^{\text{pot/dep}} = -\sum_{j \neq i} \mathbf{W}_{ij}^{\text{pot/dep}}$ imposed by hand. Thus,

$$
\frac{\partial \mathbf{W}_{ij}^{\text{F}}}{\partial \mathbf{W}_{gh}^{\text{pot/dep}}} = f^{\text{pot/dep}}\delta_{gi}(\delta_{hj} - \delta_{ij}), \qquad \frac{\partial \mathbf{q}_{ij}}{\partial \mathbf{W}_{gh}^{\text{pot/dep}}} = \pm\delta_{gi}(\delta_{hj} - \delta_{ij}).
\tag{63}
$$

The implicit $g \neq h$ that comes with all derivatives is unnecessary, as the derivatives above vanish when $g = h$.

In particular, differentiating (7),

$$\frac{\partial \mathbf{Z}_{ij}}{\partial \mathbf{W}_{gh}^{\text{pot/dep}}} = r f^{\text{pot/dep}} \mathbf{Z}_{ig} (\mathbf{Z}_{hj} - \mathbf{Z}_{gj}). \tag{64}$$

We can then differentiate expression (61) to get

$$\frac{\partial \hat{A}}{\partial \mathbf{W}_{gh}^{\text{pot/dep}}} = 2r f^{\text{pot/dep}} \mathbf{p}_g^{\infty} \left[ \sum_i a_i (\overline{\mathbf{T}}_{gi} - \overline{\mathbf{T}}_{hi}) + c_g (\eta_g^+ - \eta_h^+) \right] \pm 2 \mathbf{p}_g^{\infty} (\eta_g^+ - \eta_h^+). \tag{65}$$

where $a_i$ and $c_k$ were defined in (62).

It is sometimes useful to consider the following derivatives:

$$\frac{\partial}{\partial \mathbf{W}_{gh}^{\text{F}}} \equiv \frac{\partial}{\partial \mathbf{W}_{gh}^{\text{pot}}} + \frac{\partial}{\partial \mathbf{W}_{gh}^{\text{dep}}}, \qquad \frac{\partial}{\partial \mathbf{q}_{gh}} \equiv f^{\text{dep}} \frac{\partial}{\partial \mathbf{W}_{gh}^{\text{pot}}} - f^{\text{pot}} \frac{\partial}{\partial \mathbf{W}_{gh}^{\text{dep}}}. \tag{66}$$

Each of these derivatives behaves as their names suggest:

$$\frac{\partial \mathbf{W}_{ij}^{\text{F}}}{\partial \mathbf{W}_{gh}^{\text{F}}} = \frac{\partial \mathbf{q}_{ij}}{\partial \mathbf{q}_{gh}} = \delta_{gi} (\delta_{hj} - \delta_{ij}), \qquad \frac{\partial \mathbf{q}_{ij}}{\partial \mathbf{W}_{gh}^{\text{F}}} = \frac{\partial \mathbf{W}_{ij}^{\text{F}}}{\partial q_{gh}} = 0. \tag{67}$$

This is because we could treat $\mathbf{W}^{\text{F}}$ and $q$ as the independent variables. However, the boundaries of the allowed region are more easily expressed in terms of $\mathbf{W}^{\text{pot/dep}}$.

### 3.2.1  Scaling mode

Consider the following differential operator:

$$\Delta \equiv \sum_{g,h} \mathbf{W}_{gh}^{\text{pot}} \frac{\partial}{\partial \mathbf{W}_{gh}^{\text{pot}}} + \mathbf{W}_{gh}^{\text{dep}} \frac{\partial}{\partial \mathbf{W}_{gh}^{\text{dep}}}. \tag{68}$$

This corresponds to the scaling, $\mathbf{W}^{\text{pot/dep}} \to (1+\epsilon) \mathbf{W}^{\text{pot/dep}}$. Intuitively, this has two effects: it scales up the initial potentiation/depression and it scales down all timescales. This intuition is confirmed by the following results:

$$\begin{aligned} \Delta \mathbf{Z} &= \tau \mathbf{e} \mathbf{p}^{\infty} - \mathbf{Z}, \quad \Delta \mathbf{p}^{\infty} = 0, \quad \Delta \mathbf{T} = -\mathbf{T}, \\ \Delta \eta_i^{\pm} &= -\eta_i^{\pm}, \qquad\qquad \Delta \mathbf{q}_{ij} = \mathbf{q}_{ij}, \quad \Delta \hat{A} = 0, \end{aligned} \tag{69}$$

The anomalous bit in the scaling of $\mathbf{Z}$ is due to the lack of dependence of $\boldsymbol{\pi}$ and $\tau$ on $\mathbf{W}^{\text{pot/dep}}$.

As the area is invariant under this scaling, we can consider the $\mathbf{W}^{\text{pot/dep}}$ to be projective coordinates. Therefore we don't need to enforce the lower bound on the diagonal matrix elements while looking for the maximum area, as we can use this null-mode to enforce it afterwards without changing the area.

### 3.3  Kuhn-Tucker conditions

Consider the Lagrangian

$$\mathcal{L} = \hat{A} + \sum_{\text{pot/dep}} \sum_{i \neq j} \mu_{ij}^{\text{pot/dep}} \mathbf{W}_{ij}^{\text{pot/dep}}. \tag{70}$$

Necessary conditions for an extremum are

$$\frac{\partial \mathcal{L}}{\partial \mathbf{W}_{gh}^{\text{pot/dep}}} = 0, \qquad \mu_{gh}^{\text{pot/dep}} \geq 0, \quad \mathbf{W}_{gh}^{\text{pot/dep}} \geq 0, \quad \mu_{gh}^{\text{pot/dep}} \mathbf{W}_{gh}^{\text{pot/dep}} = 0. \tag{71}$$

with $g \neq h$. This enforces the positivity constraints on the off-diagonal elements, but not the diagonals. As discussed in §3.2.1, that can be enforced after finding the maximum using the null scaling degree of freedom.

### 3.3.1 Triangularity

Here we describe the perturbations corresponding to Figure 3(a,b) of the main paper.

Consider

$$\frac{\partial \mathcal{L}}{\partial \mathbf{q}_{gh}} = f^{\text{dep}} \frac{\partial \mathcal{L}}{\partial \mathbf{W}_{gh}^{\text{pot}}} - f^{\text{pot}} \frac{\partial \mathcal{L}}{\partial \mathbf{W}_{gh}^{\text{dep}}} = (f^{\text{dep}} \mu_{gh}^{\text{pot}} - f^{\text{pot}} \mu_{gh}^{\text{dep}}) + 2\mathbf{p}_g^{\infty}(\eta_g^+ - \eta_h^+) = 0. \qquad (72)$$

This corresponds to the shift

$$\mathbf{W}_{ij}^{\text{pot}} \to \mathbf{W}_{ij}^{\text{pot}} + f^{\text{dep}}\epsilon_{ij}, \qquad \mathbf{W}_{ij}^{\text{dep}} \to \mathbf{W}_{ij}^{\text{dep}} - f^{\text{pot}}\epsilon_{ij}, \qquad \sum_j \epsilon_{ij} = 0, \qquad (73)$$

which leaves $\mathbf{W}^{\text{F}}$ unchanged, and therefore $\mathbf{p}^{\infty}$, $\mathbf{T}$ and $\eta^{\pm}$ as well.

Assume $\eta_g^+ > \eta_h^+$. Then

$$f^{\text{dep}}\mu_{gh}^{\text{pot}} - f^{\text{pot}}\mu_{gh}^{\text{dep}} < 0 \qquad \implies \qquad \mu_{gh}^{\text{dep}} > 0 \qquad \implies \qquad \mathbf{W}_{gh}^{\text{dep}} = 0. \qquad (74)$$

Similarly, if $\eta_g^+ < \eta_h^+$, then

$$f^{\text{dep}}\mu_{gh}^{\text{pot}} - f^{\text{pot}}\mu_{gh}^{\text{dep}} > 0 \qquad \implies \qquad \mu_{gh}^{\text{pot}} > 0 \qquad \implies \qquad \mathbf{W}_{gh}^{\text{pot}} = 0. \qquad (75)$$

Thus, if we arrange the states in order of decreasing $\eta^+$, $\mathbf{W}^{\text{pot}}$ is upper-triangular and $\mathbf{W}^{\text{dep}}$ is lower triangular.

We have ignored the possibility that $\mathbf{p}_g^{\infty} = 0$, as this would imply that $\mathbf{T}_{ig} = \infty$, which would in turn imply that the Markov process is not ergodic.

### 3.3.2 Shortcuts

In this subsection we will define perturbations corresponding to Figure 3(c) of the main text.

Now consider the following combinations of derivatives for $m > 1$:

$$\widetilde{\Delta}_{g,m}^{\text{pot/dep}} \equiv \left[ \sum_{k=0}^{m-1} \frac{1}{\mathbf{p}_{g\pm k}^{\infty}} \left( \frac{\partial}{\partial \mathbf{W}_{g\pm k, g\pm(k+1)}^{\text{pot/dep}}} \right) \right] - \frac{1}{\mathbf{p}_g^{\infty}} \left( \frac{\partial}{\partial \mathbf{W}_{g,g\pm m}^{\text{pot/dep}}} \right). \qquad (76)$$

Once again, they are only well defined if all the states have non-zero equilibrium probabilities (see the comment in §3.3.1 about this being satisfied for ergodic chains).

One can show that the equilibrium probabilities, $\mathbf{p}^{\infty}$, are invariant under these operators (21):

$$\widetilde{\Delta}_{g,m}^{\text{pot/dep}} \mathbf{p}_i^{\infty} = 0, \qquad (77)$$

which makes it possible to integrate the perturbation:

$$\mathbf{W}^{\text{pot/dep}} \to \mathbf{W}^{\text{pot/dep}} + \mathbf{D}\boldsymbol{\epsilon}^{\pm(g,m)}, \qquad
\begin{aligned}
\left( \boldsymbol{\epsilon}^{\pm(g,m)} \right)_{g,g\pm m} &= -\epsilon, \\
\left( \boldsymbol{\epsilon}^{\pm(g,m)} \right)_{g\pm k, g\pm(k+1)} &= \epsilon \qquad \forall k \in [0, m-1], \\
\left( \boldsymbol{\epsilon}^{\pm(g,m)} \right)_{g\pm k, g\pm k} &= -\epsilon \qquad \forall k \in [1, m-1].
\end{aligned} \qquad (78)$$

But more interestingly for our purposes:

$$\widetilde{\Delta}_{g,m}^{\text{pot/dep}} \mathcal{L} = \left[ \sum_{k=0}^{m-1} \frac{\mu_{g\pm k, g\pm(k+1)}^{\text{pot/dep}}}{\mathbf{p}_{g\pm k}^{\infty}} - \frac{\mu_{g,g\pm m}^{\text{pot/dep}}}{\mathbf{p}_g^{\infty}} \right] + 2r f^{\text{pot/dep}} \sum_{k=0}^{m-1} \left( \eta_{g\pm k}^+ - \eta_{g\pm(k+1)}^+ \right) (c_{g\pm k} - c_g), \qquad (79)$$

In the section below, we will show that the $c_k$ are non-decreasing, if we put the states in order of decreasing $\eta_k^+$. This implies that the last term of the final expression in (79) is non-negative. If it is non-zero (there would need to be a lot of degeneracy for it to be zero), this would imply that $\mu_{g,g\pm m}^{\text{pot/dep}} > 0$, which in turn implies that $\mathbf{W}_{g,g\pm m}^{\text{pot/dep}} = 0$. This would tell us that the process with the maximal area has to have a multi-state topology.

### 3.3.3 Increasing $c_k$

In the previous subsection we defined perturbations corresponding to Figure 3(c) of the main text. In order to show that those perturbations increase the area, we must now show that the $c_k$ are non-decreasing, if we put the states in order of decreasing $\eta_k^+$.

Consider the following combinations of derivatives:

$$\Delta_{gh} \equiv \frac{1}{\mathbf{p}_g^\infty} \left( \frac{\partial}{\partial \mathbf{W}_{gh}^{\mathrm{F}}} \right) + \frac{1}{\mathbf{p}_h^\infty} \left( \frac{\partial}{\partial \mathbf{W}_{hg}^{\mathrm{F}}} \right), \tag{80}$$

$$\tag{81}$$

Note that they are only well defined if all the states have non-zero equilibrium probabilities (see the comment in §3.3.1 about this being satisfied for ergodic chains).

One can show that the equilibrium probabilities, $\mathbf{p}^\infty$, are invariant under these operators using (21):

$$\Delta_{gh} \mathbf{p}_i^\infty = 0, \tag{82}$$

which makes it possible to integrate the perturbation:

$$\mathbf{W}^{\mathrm{pot/dep}} \to \mathbf{W}^{\mathrm{pot/dep}} + \mathbf{D}\boldsymbol{\epsilon}, \qquad \begin{aligned} \boldsymbol{\epsilon} &= \boldsymbol{\epsilon}^{\mathrm{T}}, \\ \boldsymbol{\epsilon}\mathbf{e} &= 0. \end{aligned} \tag{83}$$

But more interestingly:

$$\Delta_{gh}\mathcal{L} = \frac{\mu_{gh}^{\mathrm{pot}} + \mu_{gh}^{\mathrm{dep}}}{\mathbf{p}_g^\infty} + \frac{\mu_{hg}^{\mathrm{pot}} + \mu_{hg}^{\mathrm{dep}}}{\mathbf{p}_h^\infty} + 2r\left(c_g - c_h\right)\left(\eta_g^+ - \eta_h^+\right), \tag{84}$$

$$\tag{85}$$

where $c_k$ were defined in (62).

Using the non-negativity of the Kuhn-Tucker multipliers, $\mu_{ij}^{\mathrm{pot/dep}}$, (84) tells us that if we arrange the states in order of decreasing $\eta_i^+$, the optimal process will have non-decreasing $c_k$ (if any of the $\eta_k^+$ are degenerate, we can choose their order to ensure this).

Note that, according to §3.3.1, either $\mathbf{W}_{gh}^{\mathrm{pot}}$ or $\mathbf{W}_{gh}^{\mathrm{dep}}$ will be zero at the maximum, therefore we can expect one of $\mu_{gh}^{\mathrm{pot}} + \mu_{gh}^{\mathrm{dep}}$ to be non-zero. This would rule out degeneracy of the $c_k$ or $\eta_k^+$. Looking at (72) closely, the only way $\mu_{gh}^{\mathrm{pot}} + \mu_{gh}^{\mathrm{dep}}$ could be zero is if $\eta_g^+ = \eta_h^+$ or $\mathbf{p}_g^\infty = 0$.

### 3.3.4 Summary

Using the Kuhn-Tucker formalism, we have shown that, with the states arranged in order of non-increasing $\eta_i^+$:

- There can be no ergodic maximum for which $\mathbf{W}^{\mathrm{pot}}$ contains backwards transitions or $\mathbf{W}^{\mathrm{dep}}$ contains forwards transitions.
- There can be no ergodic maximum with the $c_k$ decreasing.
- The $c_k$ may only be degenerate at an ergodic maximum if the corresponding $\eta_k^+$ are also degenerate.
- If the $c_k$ increase and the $\eta_i^+$ decrease, there can be no ergodic maximum with shortcuts.

These were shown by finding allowed perturbations that increase the area.

This leaves two possibilities for the maximum area Markov chain. Either there is no degeneracy and no shortcuts, which implies the Multi-state/serial topology that we'll discuss in §3.4, or there is some degeneracy, which would allow shortcuts provided that they do not bypass an entire degenerate set (see (79)).

Degeneracy tends to be very delicate. It is usually hard to arrange without some symmetry relating degenerate states. Such a symmetry would imply lumpability (see §1.7). The lumped chain would not have any shortcuts, as an entire degenerate set cannot be bypassed. As this lumped chain has the same area (see §2.3), we would need only consider the multi-state topology.

## 3.4 Multi-state/Serial topology

The previous results indicate that the area under the memory curve of any model is bounded by the area under the memory curve of a model with the serial/multistate topology having the same equilibrium distribution. Here we compute this area, which we will see depends only on this equilibrium distribution.

The multi-state/serial topology is defined by (see [8–10]):

$$\mathbf{W}^{\mathrm{pot}}_{ij} = q^{\mathrm{pot}}_i \delta_{i+1,j}, \qquad \mathbf{W}^{\mathrm{dep}}_{ij} = q^{\mathrm{dep}}_j \delta_{i,j+1}. \tag{86}$$

Because it has no shortcuts, it saturates various inequalities:

$$\overline{\mathbf{T}}_{ik} - \overline{\mathbf{T}}_{jk} = \begin{cases} \overline{\mathbf{T}}_{ij}, & \text{if} \quad i \le j \le k \quad \text{or} \quad i \ge j \ge k, \\ -\overline{\mathbf{T}}_{ji}, & \text{if} \quad j \le i \le k \quad \text{or} \quad j \ge i \ge k, \end{cases} \tag{87}$$

$$r\mathbf{p}^{\infty}_i \mathbf{W}^{\mathrm{F}}_{ij} \left( \overline{\mathbf{T}}_{ij} + \overline{\mathbf{T}}_{ji} \right) = 1 \quad \text{if} \quad i = j \pm 1,$$

and it satisfies detailed balance (a.k.a. reversibility a.k.a. $\mathcal{L}^2_{\mathbf{p}^{\infty}}$ self-adjointness):

$$f^{\mathrm{pot}} q^{\mathrm{pot}}_i \mathbf{p}^{\infty}_i = f^{\mathrm{dep}} q^{\mathrm{dep}}_i \mathbf{p}^{\infty}_{i+1}, \tag{88}$$

which means we can always choose the transition rates, $q^{\mathrm{pot/dep}}_i$, to give any desired equilibrium probabilities, $\mathbf{p}^{\infty}_i$.

This allows us to calculate the $c_k$'s:

$$c_k = \sum_{i<k} \mathbf{T}_{i,i+1} \left( \mathbf{p}^{\infty}_i q^{\mathrm{pot}}_i + \mathbf{p}^{\infty}_{i+1} q^{\mathrm{dep}}_i \right) - \sum_{i \ge k} \mathbf{T}_{i+1,i} \left( \mathbf{p}^{\infty}_i q^{\mathrm{pot}}_i + \mathbf{p}^{\infty}_{i+1} q^{\mathrm{dep}}_i \right),$$

$$c_{k+1} - c_k = \left( \mathbf{T}_{k,k+1} + \mathbf{T}_{k+1,k} \right) \left( \frac{\mathbf{p}^{\infty}_k \mathbf{W}^{\mathrm{F}}_{k,k+1}}{f^{\mathrm{pot}}} + \frac{\mathbf{p}^{\infty}_{k+1} \mathbf{W}^{\mathrm{F}}_{k+1,k}}{f^{\mathrm{dep}}} \right) = \frac{1}{r\, f^{\mathrm{pot}} f^{\mathrm{dep}}}, \tag{89}$$

$$\sum_k c_k \mathbf{p}^{\infty}_k = \sum_{ij} \mathbf{p}^{\infty}_i \mathbf{q}_{ij}(\eta - \eta) = 0,$$

$$\implies c_k = \frac{k - \sum_j j \mathbf{p}^{\infty}_j}{r\, f^{\mathrm{pot}} f^{\mathrm{dep}}},$$

where we used (87) to derive the first two equations respectively and Th.3 to derive the third. This allows us to write the area as

$$A = \frac{2\sqrt{N}}{r} \sum_k \left[ k - \sum_j j \mathbf{p}^{\infty}_j \right] \mathbf{p}^{\infty}_k \mathbf{w}_k = \frac{2\sqrt{N}}{r} \sum_k \left| k - \sum_j j \mathbf{p}^{\infty}_j \right| \mathbf{p}^{\infty}_k, \tag{90}$$

where we used $\mathbf{w}_k = \mathrm{sgn}(c_k)$, as discussed after (62). This reproduces equation (15) of the main paper.

In order to obtain an upper bound on the area under the memory curve of any model, we now maximise the area of the serial model with respect to its equilibrium distribution. First let us maximise (90) at fixed $\mathbf{p}^{\infty}_{\pm} = \sum_k \mathbf{p}^{\infty}_k \left( \frac{1 \pm \mathbf{w}_k}{2} \right)$. Clearly this will happen when we put all of the probability at the ends: $\mathbf{p}^{\infty}_1 = \mathbf{p}^{\infty}_-$ and $\mathbf{p}^{\infty}_n = \mathbf{p}^{\infty}_+$ are the only non-zero $\mathbf{p}^{\infty}_k$. This gives an area of

$$A \le \frac{\sqrt{N}}{r} (M - 1) \left( 4\mathbf{p}^{\infty}_+ \mathbf{p}^{\infty}_- \right). \tag{91}$$

This is maximised at $\mathbf{p}^{\infty}_+ = \mathbf{p}^{\infty}_- = \frac{1}{2}$:

$$A \le \frac{\sqrt{N}}{r} (M - 1). \tag{92}$$

This yields the area bound of equation (16) of the main text.

Note that the chain that achieves this is not ergodic, the two states at each end are absorbing. This is similar to the results found numerically in [11] in a slightly different situation.

## Footnotes

[1] Note that expanding the exponential gives