[Reviews · NeurIPS 2013]

Submitted by Assigned_Reviewer_4

This paper provides a rigorous mathematical framework for the limits of memory capacity with complex synapses. Instead of analyzing individual models with different transition matrices, the authors developed a general theory which provides an upper bound of memory capacity for the entire model space. The idea is novel and most parts of the paper are clearly written. However, whether their toy model captures sufficient amount of biological reality is questionable. I would suggest the author exploring further the connections of their theory to biology. More biological evidences are expected. They may also suggest biological experiments to test their theory. As shown in Figure 5, the numerical results are off their theoretical derivations. They should discuss more about the origin of such discrepancy. Otherwise, the theory is not convincing enough. In the mathematical derivation, the authors assume constant transition matrix between different states. It may be important to check how robust their results against perturbations of the transition matrix from time to time due to the noisy biological environment. Overall, this paper is novel and the results may provide a important guidance for the design of memory storage devices.
Summary: The author(s) developed a general theory of memory capacity with complex synapses, which bounds the functional limits of memory for the entire space of models. This is novel and interesting, however the numerical experiments do not support their theory well.

Submitted by Assigned_Reviewer_6

The paper applies the theory if ergodic Markov chains in continuous time to the analysis of the memory properties of online learning in synapses with intrinsic states extending earlier work of Abbott, Fusi and their co-workers.

The main contributions is quite interesting: a derivation of an envelope that memory curves (SNR of memory recall over time) of particular complex binary learning rules cannot cross.

Comments:

1) It would help the reader to explicitly describe the derivation of the central equation (3).

2) The original parts of the supplement should somehow be integrated in the main paper.

Summary: The paper applies the theory if ergodic Markov chains in continuous time to the analysis of the memory properties of online learning in synapses with intrinsic states.

Submitted by Assigned_Reviewer_7

The paper studies the problem of memory storage with discrete (digital) synapses. Previous work established that memory capacity can be increased by adding a cascade of (latent) states but the optimal state transition dynamics was unknown and the actual dynamics was usually hand-picked using some heuristic rules. In this paper the authors aim to derive the optimal transition dynamics for synaptic cascades. They first derive an upper bound on achievable memory capacity and show that simple models with linear chain structures can approach (achieve) this bound.

The paper is clear, high quality, generally well written and has a clear contribution to the field.

Minor comments:
line 195: Different systems in the brain might be optimised for storage at different timescale. Are there any evidence for that?
line 220: When looking at the molecular network of a synapse it is not immediately obvious what are the "states" of the system and what is the associated state transition dynamics. A strong prediction of the paper is that the molecular network in the synapse has a meaningful structure for information storage, not like the one at Fig 2a-b. Is there a simple way to check this prediction?
line 294: The authors may refer to the supplementary material around equation 14.
line 299: Isn't it problematic, than nu_i depends on M - that we try to optimize?
line 304: An intuitive description of the process depicted on Fig. 3a-b is that a potentiation has to move all synaptic states to more potentiated states, while a depression event must depress it.
line 326: In all the examples provided in the paper, half of the states are associated with w=-1 the other half has w=+1. How the memory capacity changes if the states are more asymmetric, e.g., in the most extreme case there is only a single state with w=-1 and M-1 state with w=1?
line 415: Linear chains maximize area - but not necessarily achieve the best performance (touches the envelope) at t_0? Is it true that for high performance at a certain time a chain other than linear may be optimal? Is the chain shown on Fig5b a linear chain?

I appreciate the detailed rebuttal from the authors. A short definition of the linear chain would be more useful than repeating Fig 4. A possible definition would be that a chain is linear if there is a unique ordering of the nodes from depressed to potentiated - but this is not what the authors used.
Summary: The paper is clear, high quality, generally well written and has a clear contribution to the field.
Author Feedback

Author rebuttal: We thank the reviewers for their thoughtful reviews. Below are specific comments to each reviewer.
Reviewer 1: We briefly discussed a suggestion for biological experiments to test the theory in line 428. We could expand on this. If we measured pre and post-synaptic spike trains, and also recorded changes in post-synaptic potentials to measure changes in synaptic weight, we could use hidden Markov model techniques to find the best-fit synaptic model. Then given our theory, we could match this measured synaptic model to optimal models to infer which timescales the synapse operates on. Regarding the discrepancy between the envelope and the numerical results: the envelope is just an upper bound and we do not claim that it is a tight bound at all times (see discussion). The reason is that equation (18) is not a complete set of constraints, as discussed below it. In the paragraph from line 377 to 400, we discuss this point further. Note that the numerical results are not always instructive as the numerical procedures can be prevented from reaching the global maximum by local maxima. This is shown by the fact that our hand designed models can outperform the numerical methods at late times. In fact, the apparent drop-off of the solid red curve at very late times comes from not allowing small enough epsilon. We will fix this in fig5. Yes, the level of noise tolerance is an important consideration. Note that the stochasticity of the models is an expression of biological noise. In fact, the type of noise described by the reviewer could be included by adding extra states to the model with the different transition rates. However, the synapses should not be allowed to optimize all of this noise away. It would be interesting to consider such limits on the noise levels. I think this would be beyond the scope of this work.
Reviewer 2: We'll add a derivation of equation 3 to the supplement. We could add a description along the lines of: "The factor of p^infinity describes the synapses being in the steady-state distribution before the memory is encoded. The factor of (M^pot-M^dep) comes from the encoding of the memory at t=0, with w_ideal being +/-1 in synapses that are potentiated/depotentiated. The factor of exp(rt W^F) describes the subsequent evolution of the probability distribution, averaged over all sequences of plasticity events and the factor of w indicates the readout via the synaptic weight." We referred to all of the original parts of the supplement in the main paper, but we can be more explicit, and would be happy to modify the paper to do so.
Reviewer 3: line 195: in the introduction, we cited [17], which describes diversity in synaptic structure across the vertebrate brain. This could be related to optimization for different timescales, but anything more than speculation would require a better understanding of the relation between structure and function (which is what we intend to begin with this work). line 220: Yes, fig2a,b is an example of why it is difficult to map molecular states to functional states as, despite appearances, these models actually only have two functional states due to their equivalence to fig 2c. One experimental investigation of this could be along the lines of the experiment described in line 428 (see first paragraph of our reply to reviewer 1 above for elaboration). Presumably we will find fewer functional states than molecular states. Making the link between molecular and functional states is an important research question for neurobiology. Our work helps make progress towards this by providing a theory for how functional states might be related to each other when memory is optimal, giving experimentalists clues to look for. line 294: agreed. line 299: The fact that eta_i depends on M^pot/dep is taken into account. In equation (57) of the supplement, the term involving c_g comes from this dependence. If the reviewer is concerned that the order of the eta_i could change during the maximization procedure: note that necessary conditions for a maximum only require that there is no infinitesimal perturbation that increases the area. Therefore we need only consider an infinitesimal neighborhood of the model, in which the order will not change. line 304: Yes. We'll add this phrase to the text. line 326: In fig2(a,b), we need not have an equal number of w=+/-1 states (we could change the figure to reflect this), but those states are not functionally relevant, as shown by their equivalence to fig2c. The model in fig2c, of course, has no room for such asymmetry. The models in figs4,5b do need to have equal numbers of +/- states, asymmetry would make them worse. For fig4, the effect of the asymmetry would be reduced in the limit as epsilon -> 0, as all states other than the end states would have very small p^infinity. It is important to note that while our figures do show symmetric models, our proofs are general and apply to asymmetric models as well. line 415: It is only for t_0 > sqrt(N)M that the models that nearly touch the envelope are linear chains. The model in fig5b is not a linear chain, as it has shortcut transitions, but they are only the best models (that we have found) for times t_0 < sqrt(N)M. Would it help if we repeated fig4 as fig5c?